

# An evaluation of the importance of spatial resolution in global climate and hydrological models based on the Rhine and Mississippi basins

Imme Benedict[1], Chiel C. van Heerwaarden[1], Albrecht H. Weerts[2,3], and Wilco Hazeleger[1,4]

[1]Meteorology and Air Quality Group, Wageningen University, Droevendaalsesteeg 4, 6708 BP Wageningen, The Netherlands
[2]Hydrology and Quantitative Water Management Group, Wageningen University, Droevendaalsesteeg 4, 6708 BP Wageningen, The Netherlands
[3]Deltares, P.O. Box 177, 2600 MH Delft, The Netherlands
[4]Netherlands eScience Center (NLeSC), Science Park 140, 1098 XG Amsterdam, The Netherlands

*Correspondence to:* I. Benedict (imme.benedict@wur.nl)

**Abstract.**

To study the global hydrological cycle and its response to a changing climate, we rely on global climate models (GCMs) and global hydrological models (GHMs). The spatial resolution of these models is restricted by computational resources and therefore limits the processes and level of detail that can be resolved. We assess and compare the benefits of an increased

resolution for a GCM and a GHM for two basins with long observational records; the Rhine and Mississippi basins. Increasing the resolution of a GCM ($1.125°$ to $0.25°$) results in an improved precipitation budget over the Rhine basin, attributed to a more realistic large-scale circulation. These improvements with increased resolution are not found for the Mississippi basin, possibly because precipitation is strongly depending on the representation of still unresolved convective processes. Increasing the resolution of vegetation and orography in the high resolution GHM (from $0.5°$ to $0.05°$) shows no significant differences

in discharge for both basins, likely because the hydrological processes depend highly on model parameter values that are not readily available at high resolution. Increasing the resolution of the GCM improved the simulations of the monthly averaged discharge for the Rhine, but did not improve the representation of extreme streamflow events. For the Mississippi basin, no substantial differences in precipitation and discharge were found between the two resolutions input GCM and the two resolutions GHM. These findings underline that there is no trivial route from increasing spatial resolution to a more accurately

simulated hydrological cycle at basin scale.

## 1  Introduction

Hydrometeorological extremes present a combination of atmospheric and hydrological processes. On a global scale, these processes are simulated by forcing global hydrological models (GHMs) with global climate models (GCMs). With these, we can forecast and generate future projections of the hydrological cycle and its extremes. However, the spatial resolution of

climate and hydrological models limits the details that can be resolved in a numerical simulation. With higher spatial resolution, and therefore better resolved flows and better represented landscapes, we expect more accurate results when modelling the





impact of climate on hydrological processes. However, computer capabilities are limited. Currently, the common horizontal resolution of GCMs in the Coupled Model Intercomparison Project Phase 5 (CMIP5) is around 150 km (Taylor et al., 2012b). For GHMs in the Inter-Sectoral Impact Model Intercomparison Project (ISI-MIP), this resolution is around 50 km (Haddeland et al., 2011; Schellekens et al., 2017; Beck et al., 2016).

To improve the detail level at catchment scale, it is a dilemma whether to use high resolution global models or regional downscaling. High resolution global climate models lead to better resolved large-scale processes (Scaife et al., 2011; Jung et al., 2012; Demory et al., 2014; Hodges et al., 2011), cyclones (Strachan et al., 2013; Manganello et al., 2012) and more pronounced small-scale extremes. For hydrological modelling, an increase in resolution leads to improved spatial representation of topography, soil, and vegetation (Wood et al., 2011) and therefore can result in more realistic surface runoff and evapora-

tion. However, increasing the resolution of a GHM also results in increasing unknown, and often not easily quantifiable, model parameters. This brings in large uncertainties when modelling hydrology across multiple spatial scales. There are multiple ongoing initiatives that assess the benefits of global models with very high spatial resolution for both the atmosphere (High Resolution Model Intercomparison Project; Meehl et al. 2014; Haarsma et al. 2016), and in hydrology (Wood et al., 2011; Bierkens et al., 2015).

In parallel to the research on global modelling, hydrological studies often use downscaled weather and climate variables to study regional climate variations and their hydrological impact (Jacob et al., 2014), as the spatial resolution of a basin can be substantially increased compared to a global model. Although dynamical downscaling has many benefits, it is not able to reduce biases that are related to errors in large-scale circulation patterns (Maraun et al., 2017; Van Haren et al., 2015), which are related to the low-resolution GCMs used as boundary conditions for the downscaled products (Hazeleger et al., 2015; Fowler

et al., 2007; Wood et al., 2004).

Here, we study the effect of resolution in global models on the hydrological cycle at the basin scale. The hypothesis of this study is that with higher resolution climate and hydrological models the hydrological cycle will be better simulated. We focus on two contrasting large river basins, the Rhine and Mississippi basins. To study the effect of an increased spatial resolution, we compare low- and high-resolution simulations of a global climate model, as well as of a global hydrological model over

these two basins. By comparing all cross-combinations of resolutions (Fig. 1), we aim to answer our main research question: what are the benefits of an increased resolution global climate and global hydrological model is simulating the hydrological cycle over the Rhine and Mississippi basins?

We analyse three main components of the hydrological cycle: precipitation, evaporation and discharge. We have chosen the Rhine and Mississippi basins as long measurement records are available for validation, and because their climatic drivers are

different, which can contribute to our understanding of the processes resolved with increased spatial resolution. The precipitation budget of the moderately sized Rhine basin is determined by the mid-latitude storm-track, which are shown to be better represented with higher resolution models (e.g. Davini et al., 2017a; Van Haren et al., 2015; Zappa et al., 2013). On the other hand, the precipitation budget of the Mississippi is influenced by moisture input from multiple drivers; moisture is advected from the Pacific, from the Caribbean and the Gulf of Mexico and extreme precipitation occurs within tropical cyclones (Fig. 2).

In addition, convective precipitation plays an important role over the Mississippi basin (Iorio et al., 2004). Although the Rhine


| Global climate model (GCM) | Global hydrological model (GHM) |
|---|---|
| Coarse resolution (120 x 120 km) | Coarse resolution (50 x 50 km) |
| High resolution (25 x 25 km) | High resolution (5 x 5 km) |

**Figure 1.** Two spatial resolution simulations of the GCM are used to force the GHM with two different spatial resolutions. Note that this set-up was tested for two large river basins; the Rhine and Mississippi basins.

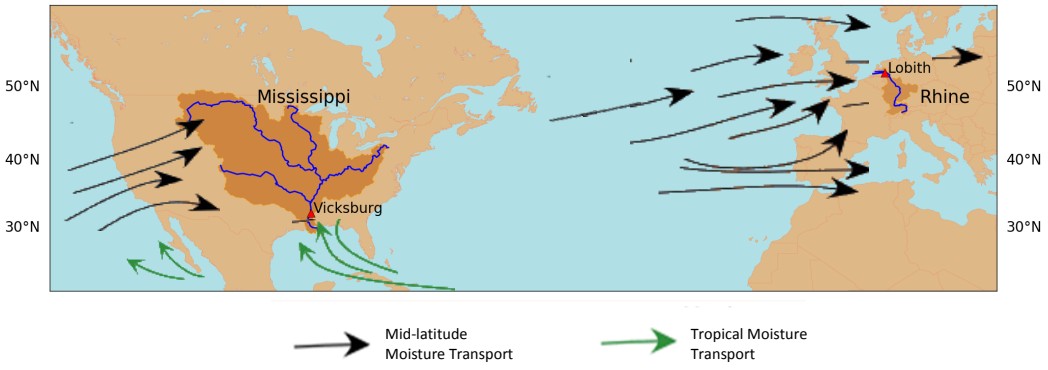

**Figure 2.** Map indicating the Rhine and Mississippi basins, the rivers, the used gauge stations (Lobith and Vicksburg) and the climatological location of mid-latitude moisture transport (black arrows) and tropical moisture transport (green arrows). Figure adapted from: http://www.physicalgeography.net/fundamentals/7s.html

and Mississippi are two contrasting basins, they do not represent the full diversity of catchments on a global scale. It would be computationally too expensive to study more regions.

The paper is structured as follows: more details about the basins is given in Sect. 2. In Sect. 3 the models, data and methods are described. We first present the results for the Rhine basin and thereafter for the Mississippi basin. The methodology of this study, as well as the broader implications, are discussed in Sect. 5 and we end with a conclusion and summary (Sect. 6).

## 2 Basin description: Rhine and Mississippi

The river Rhine originates in the Swiss Alps and flows through Switzerland, Germany and the Netherlands where it discharges into the North Sea. In this study, we analyse discharge at Lobith, which is the location where the Rhine enters the Netherlands. Therefore the basin is defined upstream of Lobith (measuring about 165 000 km$^2$, see Table 1). The average discharge at Lobith is 2 200 m$^3$ s$^{-1}$ and the highest discharges occur in late winter and spring. During this period large-scale rainfall





**Table 1.** Basin characteristics of the two study basins including basin area, used gauge station and its average discharge their.

| Basin | Basin area [km$^2$] | Gauge station | Average discharge [m$^3$ s$^{-1}$] |
|---|---|---|---|
| Rhine | 165 000 | Lobith | 2 200 |
| Mississippi | 2 981 100 | Vicksburg | 16 500 |

events, associated to storms, occur over saturated soils, which can lead to extreme flood events. Snow melt, in combination with frozen soils, can occasionally lead to extreme flood events as well (Hegnauer et al., 2014).

The Mississippi basin is more than sixteen times larger than the Rhine basin. It measures 2 981 000 km$^2$ (Table 1), which makes it the fourth-largest river basin in the world. The Mississippi River originates at Lake Itasca, Minnesota from where it

flows South towards the Gulf of Mexico. The two largest tributaries of the Mississippi are the Missouri and Ohio river. Here, we study the discharge of the Mississippi basin at Vicksburg, where the average discharge is 16 500 m$^3$ s$^{-1}$ (Table 1). Most flood events occur in winter and spring due to heavy (excess) precipitation, snowmelt and rain-on-snow events (Berghuijs et al., 2016).

## 3   Data and methodology

### 3.1   Global climate model EC-Earth

We use high resolution experiments (Haarsma et al., 2013) from the state-of-the-art atmospheric global climate model EC-Earth V2.3 (Hazeleger et al., 2010, 2012). EC-Earth is based on the European Centre for Medium-Range Weather Forecasts numerical weather prediction model Integrated Forecasting System (IFS) cy31r1. An improved hydrology scheme (H-TESSEL; Balsamo et al. 2009; Van den Hurk et al. 2000) is inserted in EC-Earth, compared to IFS. Actual evaporation is generated from

this scheme by solving the energy balance for specific land tiles. EC-Earth is forced with prescribed sea surface temperatures (SST), based on observations in current climate (NASA data at 0.25° resolution; for details we refer to Haarsma et al. 2013). Observed greenhouse gases and aerosol concentrations are also used as boundary conditions.

The high resolution experiments have a horizontal spectral resolution of T799, which corresponds to 25 km, and 91 vertical levels (further referred to as high and T799). For comparison in resolution, the same model simulations are performed with a

spectral horizontal resolution of T159, corresponding to 120 km and 62 vertical levels (further referred to as low and T159). The parameterization packages of the high and low resolution simulations are the same (Van Haren et al., 2015). The land-surface characteristics are described in the IFS model documentation (2007, IFS Documentation cy31r1, Book Chapter, ECMWF) and are interpolated to the requested resolutions (T799 and T159). For both resolutions, six members of five years (2002-2006) are created, resulting in 30 years of data representing present climate. It should be noted that the fixed boundary conditions (SST

and greenhouse forcing) decrease the independency of the members and that this research could also be performed with fewer longer simulations. More information on the experiment and the spin-up can be found in Haarsma et al. (2013).





## 3.2 Global hydrological model W3RA

W3RA is the global hydrological model that we use in this study. It is based on the landscape hydrology component model of the AWRA system (AWRA-L; Van Dijk 2010a, b). AWRA-L can be considered a hybrid between a simplified grid-based land surface model and a non-spatial, or so-called lumped, catchment model applied to individual grid cells. The model consists of two hydrological response units (HRU's); deep-rooted tall vegetation (forest) and shallow-rooted short vegetation (herbaceous), each of which occupying a fraction of a grid cell. Vertical processes are described for each HRU individually. There is no lateral redistribution of water between grid cells. This lack of lateral flow does not degrade the water balance (Van Dijk, 2010b). The model consists of three soil layers and runs with a daily time step. Actual evaporation is calculated with the energy balance. For full technical details about the model algorithm and parameters, we refer to the technical documentation (Van Dijk, 2010b). The model does not contain reservoirs.

Although W3RA is a global model, in this study we only perform the simulations for the Rhine and Mississippi basins. We run the model at the original horizontal resolution of $0.5°$ ($\sim 50$ km) and at a higher horizontal resolution of $0.05°$ ($\sim 5$ km). The parameters in W3RA at $0.5°$ resolution are determined with a regionalization approach (Van Dijk, 2010c). The list of parameters can be found in the documentation (Van Dijk, 2010b). Most of these parameters are not physically based and difficult to determine at multiple spatial scales. To allow a fair comparison between the two model resolutions, we remapped these parameters from the $0.5°$ to the $0.05°$ resolution. Our approach is verified by Melsen et al. (2016), who conclude that parameters can to a large extent be transferred across the spatial resolution (on regional scales from 1 km$^2$ to 100 km$^2$). We only make an exception for orography and vegetation, as these parameters are known at high resolution. Therefore, maps of orography and vegetation (division of HRU per grid cell) are used at the $0.05°$ resolution. The model algorithm is not adapted for the higher resolution.

The resolution of the GHM does not perfectly coincide with the resolution of the GCM (see Fig. 1). Therefore, we remap the climate variables in between using closest distance interpolation. Runoff is translated towards discharge using the wflow routing scheme (Schellekens, 2016), which is based on the kinematic wave approximation. For the $0.5°$ resolution GHM, routing is performed at $0.5°$. For the $0.05°$ resolution GHM, routing is performed at $0.083°$ as the maps of the river network are available at this resolution from the PCR-GLOBWB model (Sutanudjaja et al., 2016). We use closest distance interpolation to remap the runoff data from $0.05°$ towards $0.083°$. For each member, we perform a spin-up cycle of five years to generate the intitial conditions for the simulations of five years, from which we use the last four years for the analysis. With a soil depth of 5 m, we expect that the land-surface is in equillibrium after 6 years. When using the last four years of the simulation, hardly any effect of the initial conditions is found (results not shown). To summarize, we have 24 years of discharge simulations per combination of resolutions.

## 3.3 Observational datasets for model verification

We use the E-OBS dataset version 12.0 (Haylock et al., 2008) at $0.25°$ from 1985 until 2015 (30 years) for precipitation comparison over the Rhine basin. For extra verification, we use the genRE precipitation dataset (van Osnabrugge et al., 2017),



which is hourly data over the Rhine basin available from 1996 to 2015. For the Mississippi basin, the Climate Prediciton Center (CPC) 0.25° Daily US Unified Gauge-Based precipitation dataset version 1.0 (Higgins and Joyce., 2000) is used from 1985-2015 (30 years).

For the verification of actual evaporation, we use the GLEAM (Global Land Evaporation: the Amsterdam Methodology)
dataset version 3.0a (Martens et al., 2016) from 1985 until 2015 (30 years). This product is primarily driven by potential evaporation estimates using Priestley-Taylor (Priestley and Taylor, 1972) and by passive microwave remote sensing data.

Daily discharge data for the Rhine at Lobith and the Mississippi at Vicksburg are obtained from the Global Runoff Data Center (GRDC, 2007) from 1985 until 2015 (30 years).

In addition to the observational datasets, we verify our model results with reanalysis data from the ECMWF. A global
atmospheric reanalysis, ERA-Interim (Dee et al. 2011; further referred to as ERAI), is used from 1985 up to 2014 (30 years). ERA-Interim has a spatial resolution of around 80 km and 60 vertical levels (T255L60) and is based on IFS release Cy31r2 (comparable to Cy31r1 used in the EC-Earth simulations), which includes the land-surface TESSEL scheme (Viterbo and Beljaars, 1995). In addition, the ERA-Interim/Land reanalysis (Balsamo et al., 2013) is shortly addressed, where precipitation from ERA-Interim is corrected with satellite data and an improved land-surface scheme H-TESSEL is used (Balsamo et al.,
2009). ERA-Interim/Land is only available until 2010 and therefore we analyse the timeseries from 1985 until 2010. Lastly, the ERA20C dataset (Poli et al., 2016) is used for extra verification of the precipitation budget over the Mississippi (1985-2010). ERA20C is based on IFS cy38r1 and performs the assimilation on fewer variables than ERA-Interim.

## 3.4   Experimental set-up

We use the low and high resolution GCM EC-Earth to force the low and high resolution GHM W3RA (Fig. 1). To test the
GHM without the uncertainty of a free running GCM, we also force the GHM with ERAI data. The forcing of the GHM with the GCM is illustrated in Fig. 3. We use the following variables from the GCM: total precipitation (TP), mean sea level pressure (MSL), temperature and dewpoint temperature at 2m (T and Td), wind at 10m (U10 and V10), and surface solar and thermal radiation (SSR and STR). In the pre-process phase, potential evaporation (Epot) is calculated using Penman-Monteith (Monteith et al., 1965). Then we use potential evaporation, precipitation, temperature, mean sea level pressure and wind to
force the GHM. We do not perform a bias correction on the GCM output.

In this study, we analyse the three main components of the hydrological cycle: precipitation, evaporation and discharge. First, we analyse precipitation from the GCM, because it is the main and most uncertain forcing variable for hydrological applications (Biemans et al., 2009; Fekete et al., 2004). To get a first impression, we compare simulated and observed spatial distributions of 30-year average daily precipitation sums over the basins. Figure 2 indicates the basin areas. With the monthly
averages of basin averaged precipitation, we compare the seasonal cycle of the observations with the two resolution GCMs and ERAI. The robustness of these results are indicated with 95 % confidence intervals which are obtained after bootstrapping the daily data (Efron and Tibshirani, 1994), assuming all years to be independent. We perform an extra analysis over the Mississippi basin to better understand the precipitation patterns. We focus on the Mississippi, as extensive analysis has already



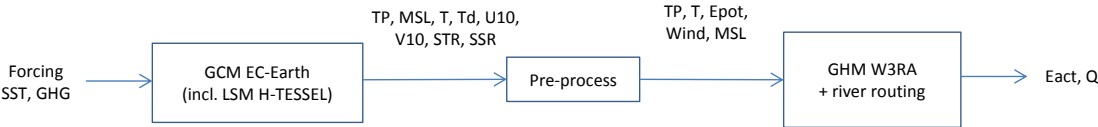

**Figure 3.** Flowchart illustrating the methodology of this study including the global climate model, the global hydrological model and the related variables: total precipitation (TP), mean sea level pressure (MSL), temperature at 2 meter (T), dewpoint temperature at 2 meter (Td), wind component x-direction at 10 meter (U10), wind component y-direction at 10 meter (V10), surface solar radiation (SSR), surface thermal radiation (STR), potential evaporation (Epot), actual evaporation (Eact) and discharge (Q).

been performed for the Rhine (Van Haren et al., 2015). We analyse the large-scale circulation patterns over the basin and we quantify the convective part of precipitation, which plays an important role in the Mississippi.

Furthermore, we statistically assess precipitation extremes by calculating the return time of annual maxima 10-day precipitation sums (van Haren et al., 2013; Shabalova et al., 2003; Kew et al., 2011). We have chosen to analyse 10-day precipitation sums, as multi-day precipitation extremes are mostly connected with extreme discharge (Disse and Engel, 2001; Ulbrich and Fink, 1995). The maxima are rank-ordered and an empirical distribution is applied to determine their return time $T$: $T = m/(N+1)$, where $m$ is the rank-ordered maxima and $N$ is the number of years in the data (30 years). Gumbel plots show the seasonal 10-day precipitation maxima as a function of the Gumbel variate $x = -\ln(-\ln(T))$, which can be translated into a return time $T$ in years. The plots are made for annual maxima in every season (DJF, MAM, JJA and SON). These Gumbel plots are only based on 30 data points, which should be taken into account during the interpretation of these plots.

Second, we analyse actual evaporation which couples the physical climate system and hydrology as it can constitute a feedback between the atmosphere and the land surface. Therefore, actual evaporation (Eact) is calculated within the global climate and global hydrological model, which allows us to compare the two models. We derive monthly averages of basin-averaged actual evaporation over the basins. We only show Eact results from the 0.5° resolution GHM.

Third, we compare monthly averaged discharge from the GHM with observations at Lobith (Rhine) and Vicksburg (Mississippi). In addition, we compare three discharge measures as defined in Table 2: $\overline{Q}_{\mathrm{mean}}, \overline{Q}_{\mathrm{max}}$ and $\overline{Q}_{\mathrm{min}}$. Finally, we determine the return times of annual maxima discharge per season, by using the same Gumbel distribution as described for precipitation. It should be noted that these results are based on 24-years of discharge simulations.

In addition, we aim to better understand the relation between precipitation and discharge. Therefore, we show scatterplots of daily discharge against previous 10-day precipitation sums for both basins, the high and low resolution GCM and the observations. For the simulations, we only show the discharge results from the 0.5° GHM but the results from the 0.05° GHM were analyzed and will be discussed where appropriate. The correlations are calculated for each season (DJF, MAM, JJA and SON) and we also include the annual maxima in discharge and 10-day precipitation sums.





**Table 2.** Four different discharge measures: $\overline{Q}_{\mathrm{mean_h}}, \overline{Q}_{\mathrm{max_h}}, \overline{Q}_{\mathrm{min_h}}$ are respectively the mean, maximum and minimum daily discharge of year number h, ranging from 1 to 24. The total number of years (H) is 24.

| Measure | Explanation | Calculation |
|---|---|---|
| $\overline{Q}_{\mathrm{mean}}$ | 24-year average mean annual discharge [m$^3$ s$^{-1}$] | $\overline{Q}_{\mathrm{mean}} = \frac{1}{H} \sum\limits_{h=1}^{24} Q_{\mathrm{mean_h}}$ |
| $\overline{Q}_{\mathrm{max}}$ | 24-year average annual maximum discharge [m$^3$ s$^{-1}$] | $\overline{Q}_{\mathrm{max}} = \frac{1}{H} \sum\limits_{h=1}^{24} Q_{\mathrm{max_h}}$ |
| $\overline{Q}_{\mathrm{min}}$ | 24-year average annual minimum discharge [m$^3$ s$^{-1}$] | $\overline{Q}_{\mathrm{min}} = \frac{1}{H} \sum\limits_{h=1}^{24} Q_{\mathrm{min_h}}$ |

All above described methods compare observations with model simulations in a statistical way. However, individual high-impact weather events, hydrometeorological extremes, are also relevant. Realistic simulations of individual events are important in forecasts, impact studies and when assessing the potential effect of antropogenic climate change. In particular, the emerging field of event attribution requires that events are plausibly simulated with numerical models (Stott et al., 2013; Hazeleger et al., 2015). In addition, single cases are often used as narratives to illustrate the complexity and linkage between components in the hydrometeorological system (Moezzi et al., 2017; Zappa and Shepherd, 2017). Therefore, the performance of this model set-up in describing hydrometeorological extremes is assessed by showing the rainfall-runoff response and synoptic pattern of a selected extreme event for each basin. This serves as an illustration on how the modelling results can be used for studying events. We show the results of the high-resolution GCM forcing the low-resolution GHM for the two basins.

# 4 Results and discussion

## 4.1 Rhine

### 4.1.1 Precipitation in the Rhine basin

The EC-Earth simulations and the observations (E-OBS and genRE) show a similar spatial distribution of precipitation over the Rhine basin (Fig. 4), with more precipitation over the Alps (4-5 mm d$^{-1}$) than downstream over Western Germany (1-2 mm d$^{-1}$). The high-resolution model shows, as expected, a more detailed distribution. A higher resolution orography reveals spatial structures such as the Alps, Ardennes and Black Forest. At the locations with large precipitation amounts, slight overestimations are found with the high resolution model (Fig. 4b). It is unclear if these overestimations are related to model performance or to underestimation of precipitation in the E-OBS dataset (Turco et al., 2013; van Osnabrugge et al., 2017), as E-OBS is based on a sparse gauge network in mountainous areas (Hofstra et al., 2009) and no correction for under catch is applied (Prein and Gobiet, 2017). The genRE precipitation dataset shows locally also higher precipitation values compared to E-OBS. Besides, it should be noted that not all Alpine, or other topographical, structures are kept within the high-resolution grid of 25 by 25 km.

From the basin-averaged precipitation sums in Fig. 5a, we find that both resolutions GCM overestimate the observed precipitation amounts. From March until July the high-resolution model outperforms the low-resolution one. Van Haren et al. (2015),





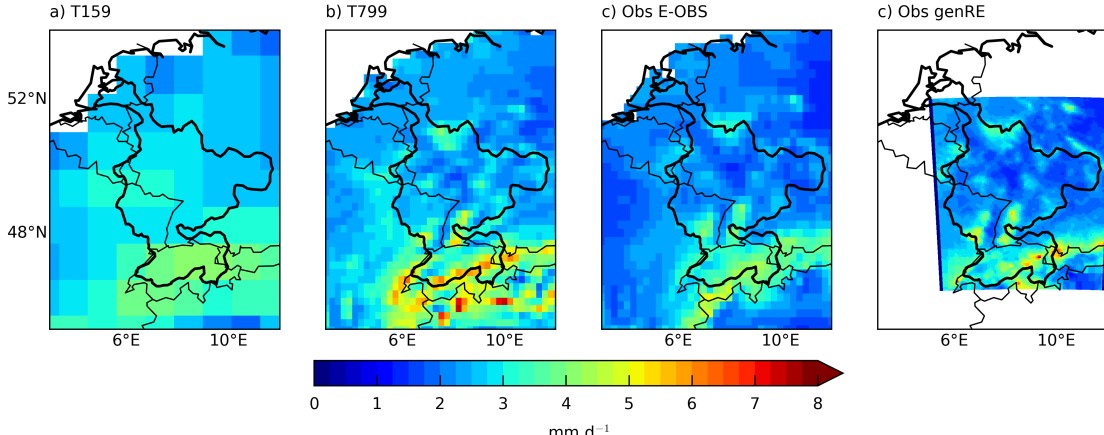

**Figure 4.** 30-year average of daily precipitation sums [mm d$^{-1}$] over the Rhine basin for a) the low resolution EC-Earth simulations (T159), b) the high resolution EC-Earth simulations (T799), c) the E-OBS dataset (Obs E-OBS) and, d) the genRE precipitation dataset (Obs genRE).

who used the same EC-Earth simulations, found similar improvements in high-resolution precipitation for the region that spans the Rhine and Meuse basin. They attributed this to the better represented storm tracks over Europe in the high-resolution simulations and therefore a more accurate horizontal moisture transport (Fig. 9 in Van Haren et al. (2015)). Nevertheless, despite the improvement with resolution, precipitation is still overestimated from January until June in T799 compared to the observations

and ERAI (Fig. 5a).

Figure 6 (left panels) shows the influence of resolution on the return time of annual 10-day precipitation maxima per season. During all seasons, and particularly in DJF and MAM, there is a distinct overestimation of precipitation by EC-Earth at lower return times (smaller than two years). This is in agreement with the overestimation in the monthly averages of precipitation (Fig. 5a). At higher return times (larger than two years), we find an underestimation of precipitation in the GCM data in DJF

(Fig. 6a). The extremes in the storm-track season (SON) are quite well reproduced by the model. By comparing the two model resolutions, we find that in MAM and JJA the high-resolution model outperforms the low-resolution one for all return times, which suggests that with an increased resolution the right large-scale conditions are present to activate convection.

### 4.1.2   Actual evaporation in the Rhine basin

In Fig. 5b we show actual evaporation from GLEAM, EC-Earth, ERAI and the 0.5° GHM forced with EC-Earth and ERAI.

Actual evaporation (Eact) is overestimated in all simulations compared to the reference GLEAM, especially in winter (0.5 mm d$^{-1}$). This can be related to an overestimation of precipitation in winter, as an increase in precipitation can lead to larger evaporation rates. Actual evaporation from the high-resolution shows a smaller bias with observations than the low-resolution,



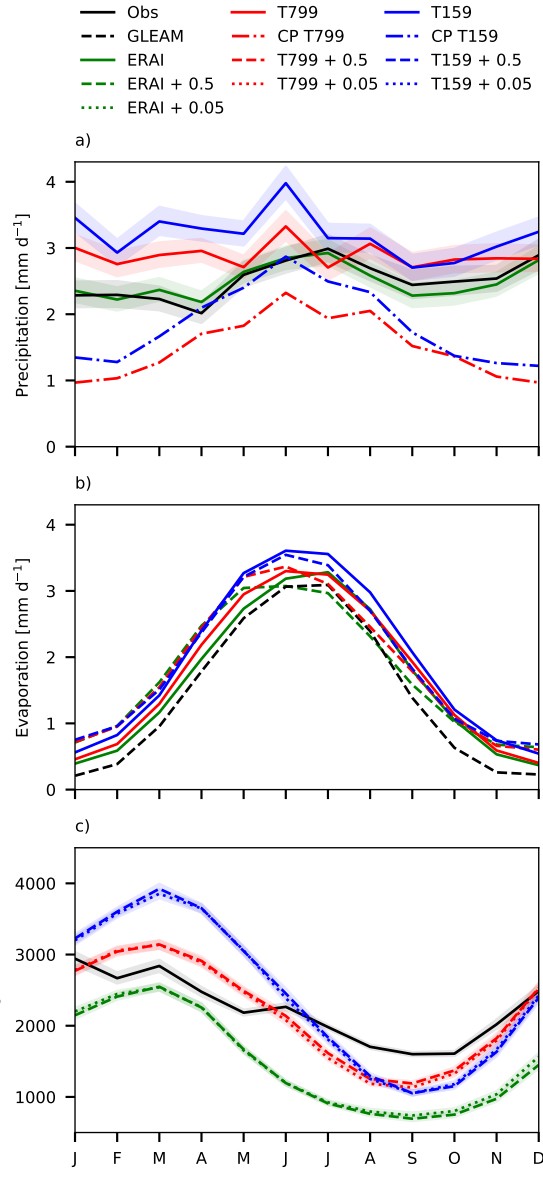

**Figure 5.** Monthly averages of a) basin-averaged daily precipitation sums [mm d$^{-1}$], b) basin-averaged daily evaporation sums [mm d$^{-1}$] and c) daily discharge [m$^3$ s$^{-1}$] at Lobith for the Rhine basin. Black lines are observations, green is ERAI. The red and blue lines are respectively the high resolution (T799) and low resolution (T159) GCM. The dash-dotted lines indicate convective precipitation, dashed lines output from the 0.5° GHM and dotted lines from the 0.05° GHM. The shaded bands indicate the 95 % confidence intervals.



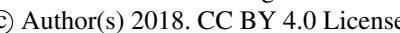

**Figure 6.** Gumbel plots of seasonal (DJF, MAM, JJA and SON) maximum 10-day precipitation sums [mm] over the Rhine (left panels) and maximum discharge [m³ s⁻¹] at Lobith (right panels) and their related return times T expressed in standardized Gumbel variate $x = -\ln(-\ln(T))$. Observed discharges are shown in black, high resolution forcing (T799) in red, low resolution forcing (T159) in blue and forcing with ERA-Interim in green. The discharge results are output from the 0.5° GHM.




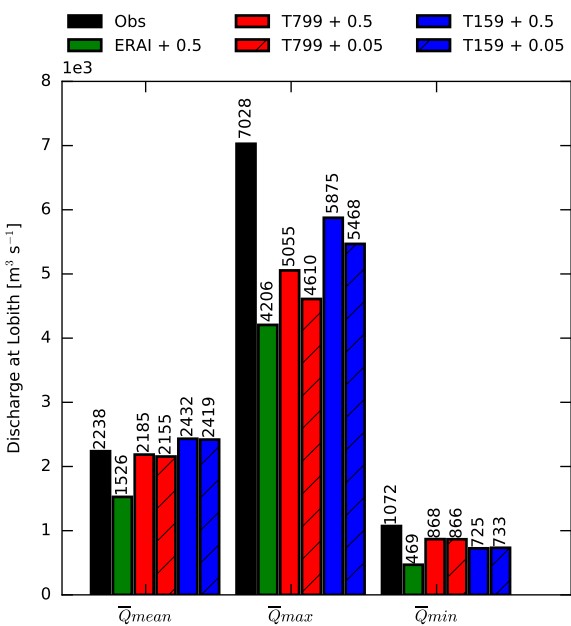

**Figure 7.** $\overline{Q}_{\mathrm{mean}}, \overline{Q}_{\mathrm{max}}$ and $\overline{Q}_{\mathrm{min}}$ in m$^3$ s$^{-1}$ at Lobith for the observations and the different combinations of simulations.

which is consistent with our precipitation results. We also find an overestimation of Eact from ERAI compared to GLEAM (Fig. 5b), though precipitation in ERAI is not overestimated (Fig. 5a). These high evaporation amounts in ERAI can explain the large underestimation of simulated discharge at Lobith, discussed in the next Sect. 4.1.3.

There is a large difference between actual evaporation directly from ERAI and actual evaporation indirectly from the GHM

forced with ERAI. This difference is smaller for the EC-Earth simulations. Possibly, this is because of an improved land-surface scheme in EC-Earth (H-TESSEL), while ERAI is based on the old scheme (TESSEL) that does not contain a seasonal cycle in leaf area index and has a global uniform soil texture (Balsamo et al., 2009).

The yearly-averaged Eact from the climate and hydrological model are comparable, but there are seasonal differences (Fig. 5b). As both models (GCM and GHM) solve actual evaporation from the energy balance, these differences are related to the

vegetation and soil characteristics of the models. Actual evaporation from the GHM is higher in the beginning of the year (January until June) and peaks earlier in the season compared to the GCM (Fig. 5b). Overall, it seems that the Eact from the GCM is in better agreement with the reference GLEAM dataset.

### 4.1.3  Discharge in the Rhine

In Fig. 5c we show monthly-averaged discharge at Lobith from the 0.5° and 0.05° GHM, forced with EC-Earth T799, EC-Earth

T159 and ERAI. Observed discharge is also shown. Figure 7 shows the discharge measures in a barplot.



The discharge simulated with ERAI forcing largely underestimates the observed discharge ($\sim 700$ m$^3$ s$^{-1}$), in particular from June until December (Fig. 7 & 5c). Photiadou et al. (2011) and Szczypta et al. (2012) present similar results, which they relate to an underestimation of precipitation in ERAI (Balsamo et al., 2010). However, our results show good estimates of basin-averaged precipitation from ERAI, except for a slight underestimation from August to November (Fig. 5a). Therefore,

we conclude that the GHM is too dry in the summer months for the Rhine basin, introducing a negative bias in discharge. We also find lower discharges in the end of summer with EC-Earth forcing, possibly related to the dry bias of the GHM. From February until May, the overestimations in precipitation from both resolutions GCM are reflected in overestimations of discharge, with the largest bias for the low-resolution forcing (Fig. 5c & $\overline{Q}_{\max}$ in Fig. 7).

For the discharge extremes, we show similar Gumbel plots as for precipitation, now for annual maxima discharges per

season (right panels in Fig. 6). The differences found in the return times of 10-day precipitation sums between the high and low resolution simulations are reflected in the differences found in the return values for the discharge, in every season. However, the differences between simulations and observations are not consistent from precipitation to discharge. Firstly, this is because the hydrological model has a large influence on the discharge results, which was already seen from the monthly average discharge plots. For example, the dry bias of the model results in lower discharge extremes in SON (Fig. 6h). Secondly, there is not a

one-to-one correlation between precipitation sums and discharge, as is shown more extensively in the next Sect. 4.1.4. Lastly, the river Rhine is highly regulated which affects the observations but not the simulations.

Overall, we can conclude that with the high-resolution EC-Earth forcing the seasonal cycle and the monthly averaged discharges are better represented compared to the low-resolution forcing, mainly because of improvements in precipitation. The difference in precipitation between the model resolutions is clearly reflected in discharge, although biases in the hydrological

model also influence these results. The discharge extremes ($\overline{Q}_{\min}$ and $\overline{Q}_{\max}$) are not consistently improved with high-resolution forcing. It is not clear from these analysis if that is related to the forcing or to the performance of the hydrological model. The results are robust based on our modelling system.

We also tested the resolution sensitivity of the global hydrological model. We find small, but not significant differences in the discharge (measures) between the 0.5° and 0.05° model; the high resolution GHM (0.05°) gives slightly lower annual mean

discharge results. With the 0.05° model, the peak flows are less extreme and the low-flows are similar to the low-flows from the 0.5° model. Because of a higher resolution orography, a more detailed river network is present in the high resolution model. Due to the presence of extra tributaries the response of precipitation to the main river may be damped, leading to a decrease of the peak flow.

### 4.1.4 Outlook on the extremes for the Rhine

In previous sections, we showed that, compared to observations, the mean (monthly) statistics of precipitation, actual evaporation and discharge are improved with high-resolution modelling. We show the correlation between 10-day precipitation sums and daily discharge in Fig. 8. It should be noted that by applying a moving window over the 24-year timeseries, individual events are reflected in multiple subsequent data points. We find highest correlations during winter and lowest correlations during summer. In summer, more evaporation occurs which decreases the correlation between precipitation and discharge. In





winter, precipitation amounts are large and there is almost no evaporation, leading to higher discharges. In spring (MAM), fast surface runoff can be generated by rain occuring over saturated soils and rain-on-snow events (McCabe et al., 2007). We also find that the difference in correlations between the seasons is better represent in the high-resolution than the low-resolution, compared to observations (Fig. 8). In addition, the distribution of discharge and precipitation values of the high-resolution

forcing compares better to observations. The low discharge values, which occur in JJA with the low-resolution forcing, can be related to two events in two members of the simulations, and are unrealistic. The correlations of precipitation and discharge from the high-resolution GHM are not shown here, as these distributions are similar to the distributions with the low-resolution GHM (Fig. 8a and b), except that less high peak flows are found with the higher resolution model (Fig. 7).

To illustrate how the models simulate a high-impact event, we show here an event for the Rhine basin. The selected event
is indicated with an open circle in Fig. 8b and is an annual maximum in precipitation, occurring at the end of November. The average precipitation sum in SON is 30 mm. In this case the sum is 103 mm, resulting in a discharge of almost 9 000 m$^3$ s$^{-1}$. Figure 9a shows the rainfall-runoff distribution from 20 days before, until 10 days after the selected event. In addition, the synoptic situation is shown with 10-day averaged mean sea level pressure, vertical integrated moisture fluxes and the 10-day precipitation sums (Fig. 9b). From the mean sea level pressure and moisture fluxes, we can infer that there is a low pressure
system (mid-latitude cyclone) situated over the North Atlantic, before the coast of Norway, bringing moisture from the Atlantic over Europe leading to extreme precipitation over the Alps.

This single case is an example of the linkage between components in the hydrometeorological system; large-scale circulation associated with extreme precipitation and high discharges for the Rhine basin. In this case, the high-resolution GCM is able to simulate patterns that better correspond to observations. This does not mean that the low-resolution GCM is not able to
simulate such circulation patterns but previous studies have shown common biases among low-resolution GCMs, such as a too zonal storm-track (Chang et al., 2012; Van Haren et al., 2015; Zappa et al., 2013).

## 4.2 Mississippi

### 4.2.1 Precipitation in the Mississippi basin

While precipitation over the Rhine is dominated by the storm-track, the Mississippi basin has multiple climatic drivers (Fig.
2). Moisture is advected from the Pacific resulting in high precipitation amounts over the Rocky mountains (4-5 mm d$^{-1}$). The Great Plains, which are situated on the lee side of the Rockies, are relatively dry (1-2 mm d$^{-1}$), whereas the South-East of the USA is relatively wet (3-4 mm d$^{-1}$) because of convection and advection of moisture from the warm tropical Caribbean and Gulf of Mexico. Figure 10 shows the distribution of seasonal averaged precipitation over the Mississippi basin for the two resolutions of the GCM and the observations (CPC). There are clear improvements in the distribution of precipitation for
the high resolution GCM over mountain ranges attributed to better representation of orography, such as over the Rockies, the Cascades, and the Sierra Nevada, which is in line with previous resolution studies with an atmosphere-only GCM (Duffy et al., 2003) and a coupled ocean-atmosphere GCM (van der Wiel et al., 2016). Comparison of the simulations with observations reveals an overestimation of precipitation in the North-East of the catchment in DJF and MAM (Fig. 10). In SON, in the South





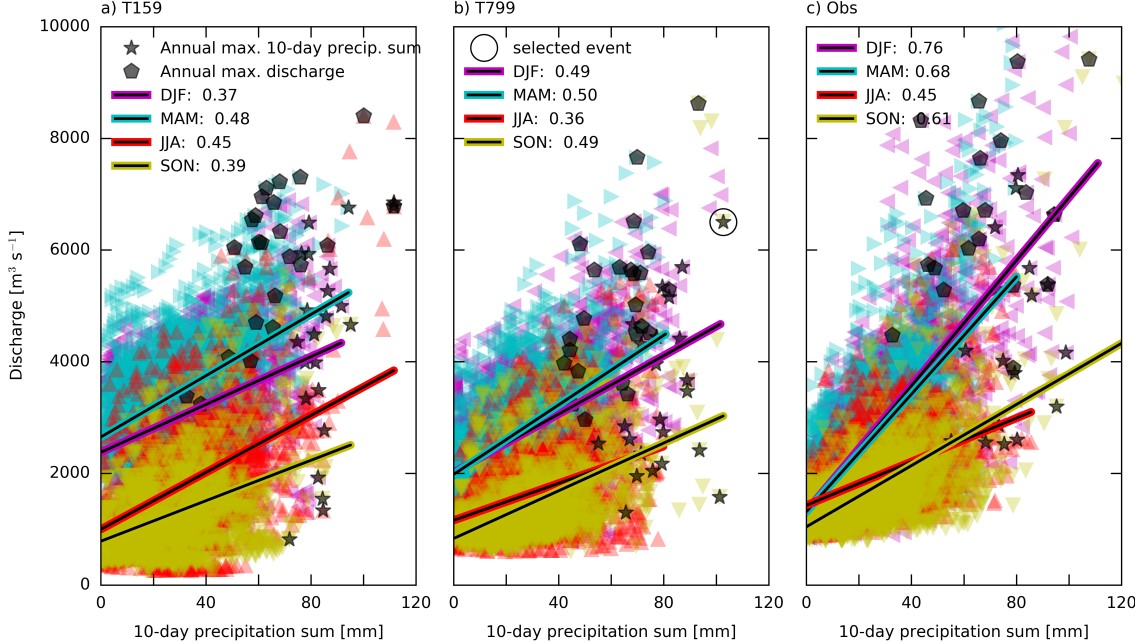

**Figure 8.** Scatterplot for the Rhine basin of daily discharge [m³ s⁻¹] with previous 10-day precipitation sums [mm] for a) the low resolution forcing (T159), b) the high resolution forcing (T799) and c) the observations (Obs.). The discharge results shown here are obtained with the 0.5° GHM. The different seasons are indicated with the colours and regression line and correlation value. The annual maxima of both 10-day precipitation sums and discharge are indicated with respectively the black stars and hexagons.

of the Mississippi basin, the high-resolution model shows higher precipitation amounts, comparable to the observations. These are not found in the low-resolution model. This could possibly indicate that cyclones are better captured in the high resolution model which bring precipitation along the coast.

Monthly and basin averaged daily precipitation sums of both simulations show a shift of one to two months in the seasonal
5   cycle, where the highest monthly values occur in April/May instead of in June (Fig. 11a). Moreover, the amount of precipitation in this shifted peak is overestimated (Fig. 11a). The increase in precipitation in October-November is not observed but occurs, most pronounced, in the high-resolution simulations. A similar peak in October-November is found in the convective part and suggests a bias in convection in the high-resolution model. Similar precipitation biases are found in the EC-Earth simulations for the sub-basin averages (Missouri and Arkansas-Red, not shown). In contrast to the EC-Earth simulations, precipitation
10   from ERAI shows the correct seasonal cycle (Fig. 11a). EC-Earth and ERAI are based on the same atmospheric model (IFS), albeit different versions. Therefore we hypothesize that the precipitation bias found with EC-Earth, is not present in the ERAI reanalysis, because of the data assimilation process. The precipitation budget from the ERA20C reanalysis data, where assimilation is performed on fewer variables than ERAI, shows a larger bias with observations compared to ERAI, supporting our hypothesis (ERA20C data not shown).





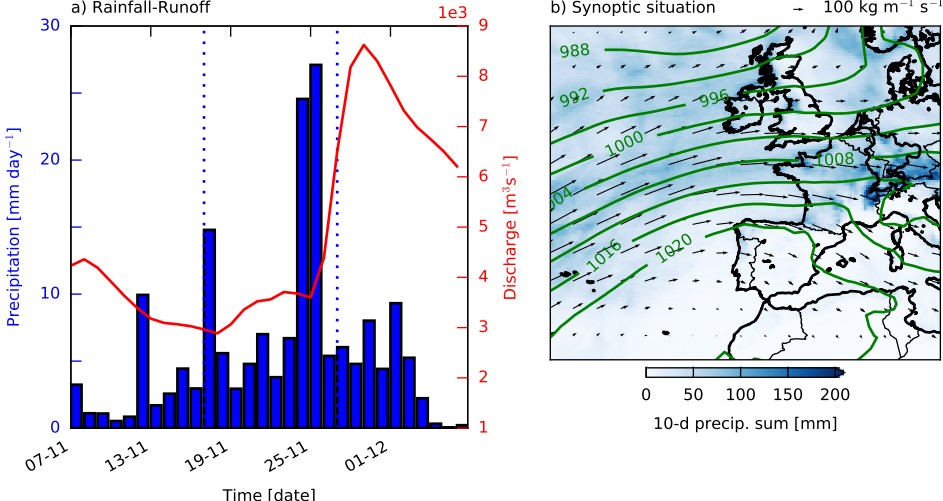

**Figure 9.** In a) precipitation (in blue) and discharge (in red) for the Rhine are shown 20 days before and 10 days after the selected event. The vertical dotted lines indicate the 10 day period, which is spatially summed in b). The contour lines in b) indicate the 10-day averaged mean sea level pressure in hPa and the arrows the 10-day averaged vertical integrated moisture fluxes in kg m$^{-1}$ s$^{-1}$.

Apart from the precipitation bias between EC-Earth simulations and observations, no substantial differences in basin-averaged precipitation between the low and high resolution simulations were found (Fig. 11a). This similarity between the two resolutions GCM could be explained by the convective component of precipitation, which is modelled at the sub-grid scale (i.e. parameterized) for both resolutions. We will further discuss convection in the next Sect. (4.2.2). Thereby, we will also

assess the sensitivity of resolution to the large-scale circulation over the Mississippi basin.

The bias between observations and simulations is also reflected in the Gumbel plots of 10-day precipitation sums per season over the basin (left panels Fig. 12). In MAM, there is an overestimation of the extremes for all the return times and in JJA an underestimation for all the return times. In SON, there are much larger precipitation extremes in the high resolution compared to the low resolution (Fig. 12). This could possibly be related to the improved simulation of tropical cyclones with higher

resolution, although this should be investigated further. In DJF, we find larger biases with the high-resolution compared to the low-resolution, although previous studies show improvements of extreme precipitation with increased resolution (Iorio et al., 2004; Wehner et al., 2010; van der Wiel et al., 2016; Duffy et al., 2003). In the winter season moisture advection from the Pacific plays a large role. A more detailed orography in the high-resolution simulations could trigger more precipitation leading to overestimations. In addition, 'observed' precipitation products, like the CPC dataset, severely underestimate precipitation

over the western mountain ranges (Lundquist et al., 2015; Henn et al., 2017).



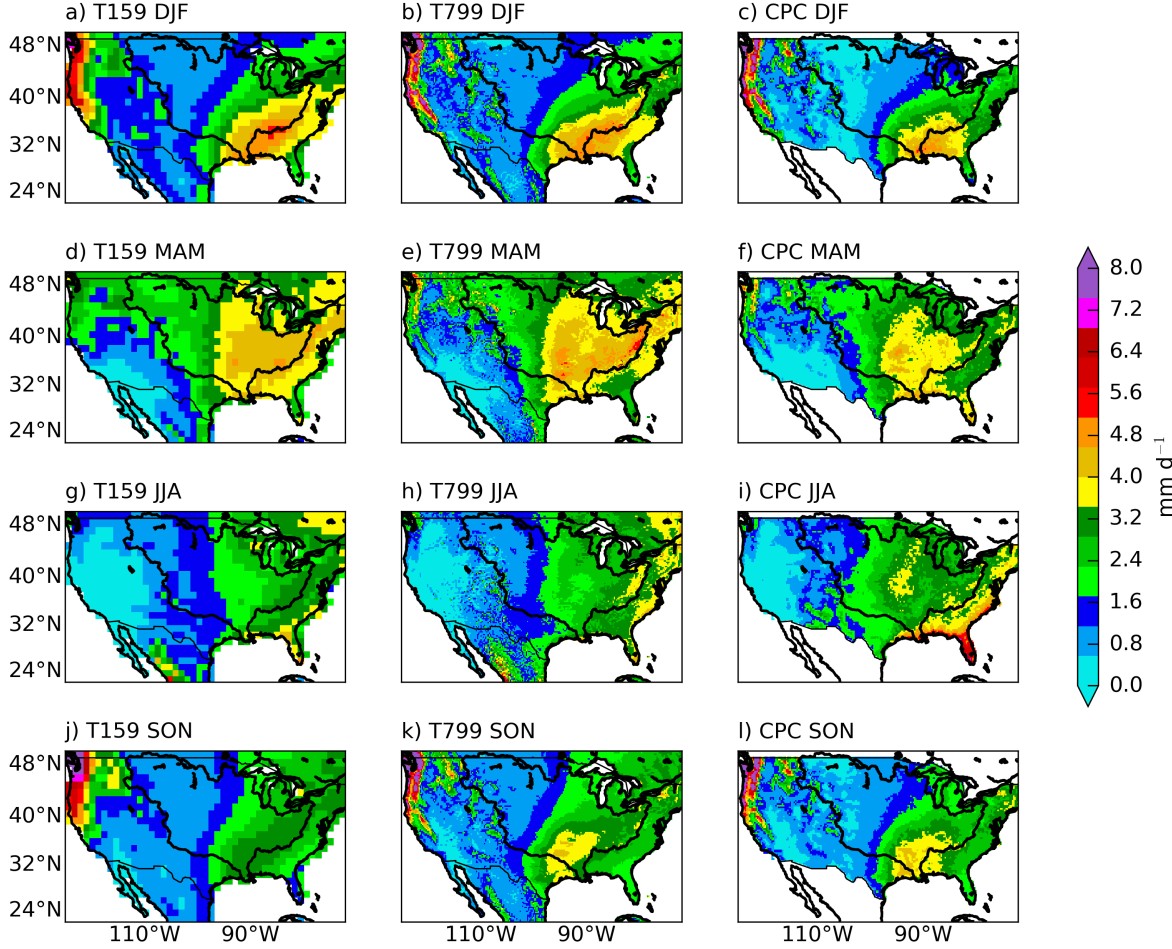

**Figure 10.** Seasonal means (DJF, MAM, JJA and SON) of daily precipitation sums[mm d$^{-1}$] from the low resolution EC-Earth simulations (T159, left columns) the high resolution EC-Earth simulations (T799, middle columns) and the observations (CPC, right columns).





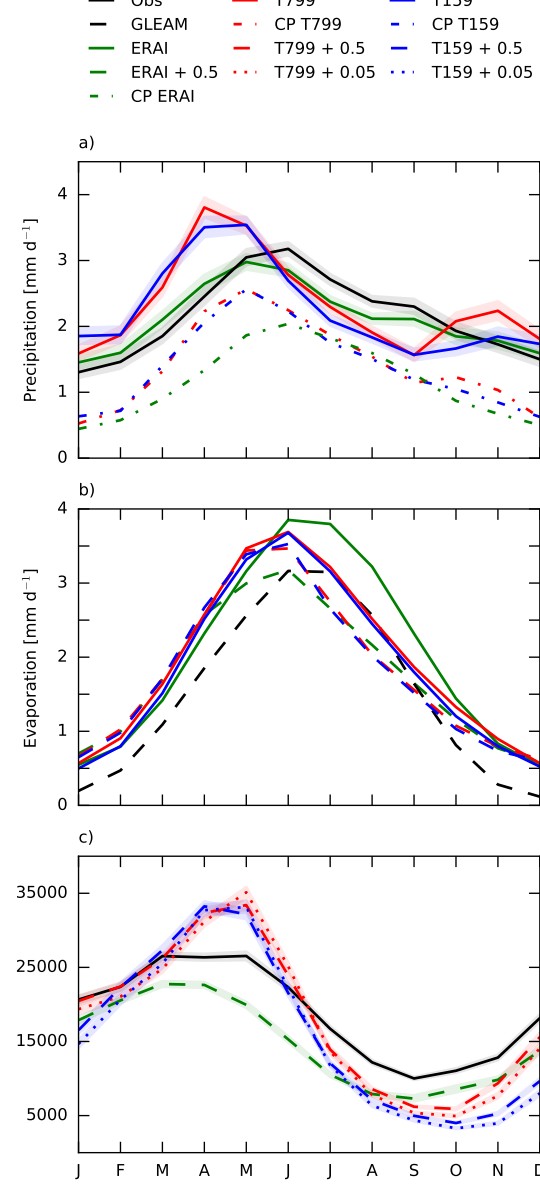

**Figure 11.** Monthly averages of a) basin-averaged daily precipitation sums [mm d$^{-1}$], b) basin-averaged daily evaporation sums [mm d$^{-1}$] and c) daily discharge [m$^3$ s$^{-1}$] at Vicksburg for the Mississippi basin. Black lines are observations, green is ERAI. The red and blue lines are respectively the high resolution (T799) and low resolution (T159) GCM. The dash-dotted lines indicate convective precipitation, dashed lines output from the 0.5° GHM and dotted lines from the 0.05° GHM. The shaded bands indicate the 95 % confidence intervals.







**Figure 12.** Gumbel plots of seasonal (DJF, MAM, JJA and SON) maximum 10-day precipitation sums [mm] over the Mississippi (left panels) and maximum discharge [m³ s⁻¹] at Vicksburg (right panels) and their related return times T expressed in standardized Gumbel variate $x = -\ln(-\ln(T))$. Observed discharges are shown in black, high resolution forcing (T799) in red, low resolution forcing (T159) in blue and forcing with ERA-Interim in green. The discharge results are output from the 0.5° GHM.



### 4.2.2 Resolution analysis of the Mississippi basin

In the previous Sect. 4.2.1, our results show that a bias exist between simulated and observed basin-averaged precipitation for the Mississippi, especially in MAM ($\sim$ 0.5-1 mm d$^{-1}$, Fig. 11a). Moreover, no substantial differences in precipitation are found between the low and high resolution simulations, except for SON (Fig. 11a). This is in contrast with our results

for the Rhine basin, where better precipitation estimates are found with the high resolution GCM, because of better resolved large-scale circulation patterns (Van Haren et al., 2015). Here, we will shortly assess the resolution sensitivity of large-scale circulation and the role of convection over the Mississippi basin.

We show the precipitation generated by the convective parameterization as monthly averages in Fig. 11a. The monthly averages of convective precipitation are very similar for the two resolutions GCM. Convective precipitation from ERAI shows

a different seasonal cycle, with a peak later in the season (Fig. 11a). This suggest that the bias in total precipitation in EC-Earth is mainly related to a bias in convective precipitation. The large contribution of convective precipitation to total precipitation in the model likely explains why we do not find differences in basin-averaged precipitation between the two resolutions in MAM and summer (Fig. 11a), as convective cloud systems are smaller than both model resolutions grid size and therefore parameterized. This is confirmed by Iorio et al. (2004) who found no improvements in precipitation over the USA in MAM and

JJA with increased resolution, which was related to the dominance of convective precipitation in these two seasons. Balsamo et al. (2010) mentioned that large-scale weather systems in winter are easier to simulate in numerical weather predictions than convective systems in summer. There are also studies which show that the link between soil moisture and precipitation is incorrect in models that parameterize convection (Hohenegger et al., 2009; Taylor et al., 2012a). Recently, convection-permitting simulations over the USA were perfomed (Liu et al., 2017), which show good performance capturing the seaonal

precipitation climatology, except for a dry bias in summer. In addition, the main characteristics of mesoscale convective systems were well captured in these new simulations (Prein et al., 2017).

Besides convection, large-scale structures bring moisture from the Pacific over the Rockies and from the Caribbean and Gulf of Mexico with the low level jet to the Mississippi. The resolution dependency of these large-scale processes is assessed by analysing geopotential height at 500 hPa (data not shown) and 850 hPa (Fig. 13). We find that these patterns are very similar

between EC-Earth T799, EC-Earth T159 and ERAI. In addition, we also show moisture convergence, as defined under steady state: $P - E = -\frac{1}{g} \nabla \cdot \int_0^{p_s} Vq dp$ , where P is precipitation, E is evaporation, g is the gravitational constant, V represents the horizontal wind components and q is specific humidity. We define moisture convergence positive and derive it from evaporation and precipitation. The overall patterns of moisture convergence are similar for EC-Earth T799, EC-Earth T159 and ERAI. Differences on the local scale can be related to differences in resolution and therefore the representation of orography. From

the difference plots (Fig. 13 d & e) we find that the moisture convergence is more similar between the two resolutions EC-Earth than between the high-resolution EC-Earth and ERAI. This is in line with the precipitation patterns we found (Fig. 11a), which are similar between the two resolutions but quite different in ERAI. There is more convergence in the high-resolution GCM compared to ERAI, which also results in more precipitation in the high-resolution GCM. There is also slightly more





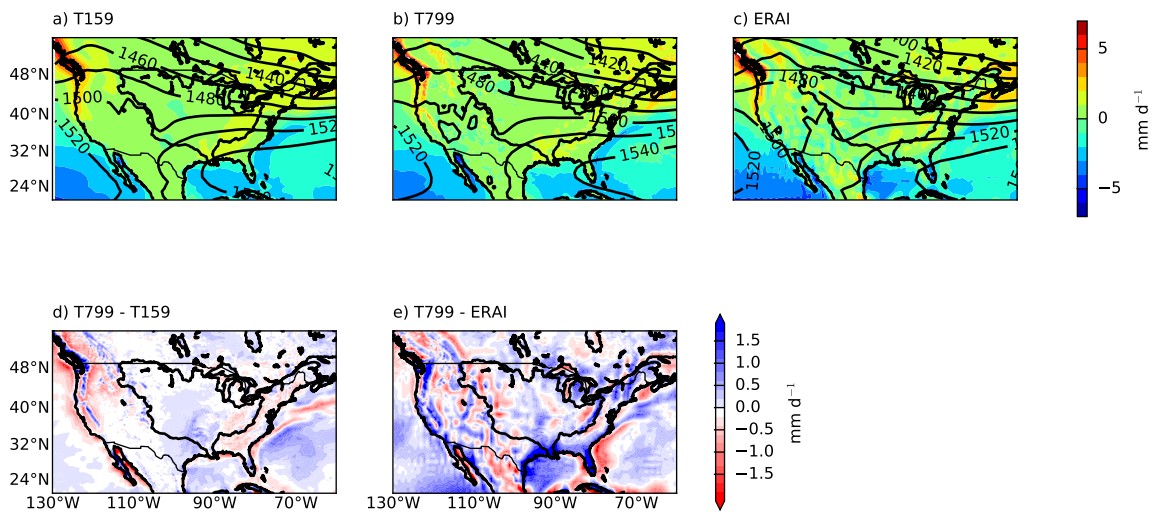

**Figure 13.** 30 year averages of geopotential height [m] at 850 hPa and moisture convergence [mm d$^{-1}$] over the Mississippi basin for a) the low-resolution GCM (T159), b) high-resolution GCM (T799), c) ERAI, and the difference between d) high and low resolution GCM (T799-T159) and e) high resolution GCM and ERAI (T799-ERAI).

convergence in the high resolution EC-Earth compared to the low resolution, and we also found slightly higher monthly average precipitation in SON.

To summarize, this resolution analysis suggests that the positive bias in precipitation in EC-Earth is mainly related to the convective part of precipitation. A first analysis of the geopotential fields (500 and 850 hPa) shows that the large-scale patterns are very similar between the resolutions of EC-Earth and ERAI. We do find that the difference in moisture convergence between the two resolutions GCM is smaller than between the GCM and ERAI. This possibly indicates that the triggering of convection is different between the GCM and ERAI. However, we recommend further analysis to confirm these results.

### 4.2.3 Actual evaporation in the Mississippi basin

A consistent pattern between evaporation and precipitation is found in the simulations for the Mississippi basin. The shift in seasonal cycle in the EC-Earth precipitation budget is reflected in a similar shift in the Eact budget (Fig. 11b). Furthermore, there are no substantial differences found in Eact between the two resolutions GCM. Nevertheless, we find large overestimations ($\sim$ 0.5 mm d$^{-1}$) of Eact in winter (NDJF) in the simulations compared to the GLEAM dataset. In November and December, these overestimations can not be related to the precipitation budget. These high amounts of evaporation in winter are also found for the Rhine and are therefore possibly related to the performance of the GHM.



The largest overestimations of actual evaporation are from the ERAI data, which was also shown by (Betts et al., 2009). The land-surface scheme of ERAI (TESSEL) has a fixed leaf area index (van den Hurk et al., 2003) and a global uniform soil texture leading to low amounts of surface runoff (Balsamo et al., 2009), which could induce smaller amounts of interception and open water evaporation resulting in overestimations of evaporation. Moreover, there are large differences in actual evaporation from

ERAI directly and from the GHM forced with ERAI (Fig. 11b). These differences are larger for ERAI than for EC-Earth, which was also observed for the Rhine basin.

The actual evaporation from the GHM decreases faster from June onwards compared to the actual evaporation from the GCM. A similar sudden decrease was found in the discharge at Vicksburg. In other words, there occurs a quick drying in the GHM from May-June. This should be mainly related to the vegetation and soil characteristics of the GHM, as the GCM does

not show the quick drying. Overall, it is hard to judge whether the evaporation product from the GCM or the GHM performs better in comparison with the observations as the seasonal bias in precipitation is also influencing the evaporation budget.

### 4.2.4 Discharge in the Mississippi

We show the monthly averaged discharge at Vicksburg in Fig. 11c and the different discharge measures in Fig. 14. We find an underestimation of the ERAI forced discharge during the whole year compared to the observed discharge. We can only partly

explain this with the underestimation of ERAI precipitation in JJA (Fig. 11a). Precipitation from the ERAI/Land product agrees very well with the observations, however, discharge is still underestimated (data not shown). Therefore, we conclude that most of the underestimation in discharge is related to an overestimation of actual evaporation, which was shown in Sect. 4.2.3.

Annual mean discharge is underestimated ($\sim$ 2 000 m$^3$ s$^{-1}$) with the low-resolution forcing and well simulated with the high-resolution forcing ($\overline{Q}_{\mathrm{mean}}$ in Fig. 14).The monthly-averaged discharge forced with EC-Earth is too high in spring, because

of too high precipitation values (Fig. 11). In January and February, precipitation (including snow) is also overestimated in EC-Earth leading to increased discharges in April and May when the temperature rises. From May onwards the discharge decreases more rapidly in the model than observed. During the rest of the year, there is a clear discharge response to the precipitation budget. It is possible that in October-November the improvements in discharge for the high-resolution exist for the wrong reason, as the second precipitation peak in the high-resolution is not seen in the observations.

For the extremes in SON, we also find a clear difference between the high and low resolution forcing (Fig. 12g & h). With high-resolution forcing larger extremes are found, although discharge is still underestimated for lower return values, which was also found for the monthly averages. In DJF, there is a clear difference between the two resolutions for the largest return values in precipitation and this is also reflected in the return values for discharge, which are larger with high resolution forcing. In MAM, precipitation (extremes) are largely overestimated in EC-Earth, which is reflected in slight overestimations of discharge

in the lower return values but large overestimations for the higher return values (Fig. 12 c & d). As the GHM does not take into account reservoirs, a faster response of discharge on precipitation in the model simulations is expected compared to the observations. In the summer months (JJA), the discharge extremes are quite well represented by the model. Nevertheless, the ERAI forced discharge underestimates the extremes in these months.





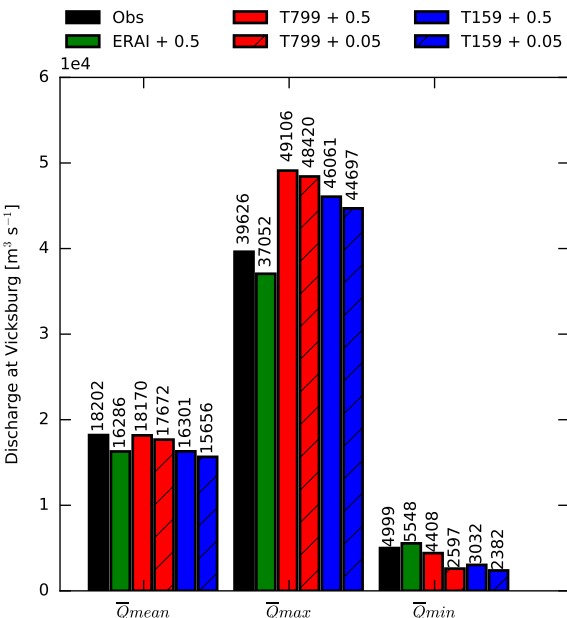

**Figure 14.** $\overline{Q}_{\mathrm{mean}}, \overline{Q}_{\mathrm{max}}$ and $\overline{Q}_{\mathrm{min}}$ in m$^3$ s$^{-1}$ at Vicksburg for the observations and the different combinations of simulations.

In general, for the monthly averages and lower return values, the dry bias of the GHM is clearly reflected in the results. For the extremes with higher return values, we find that the signal of the precipitation extremes is reflected in the discharge extremes and the model performance plays a less important role. There are no substantial differences in discharge between the 0.5° and 0.05° resolutions, as was also found for the Rhine.

**4.2.5 Outlook on the extremes for the Mississippi**

Figure 15 shows the correlations between 10-day precipitation sums and discharge for the two resolution simulations and the observations over the Mississippi basin. For the simulations (Fig. 15 a & b), we find highest correlations in summer and lowest correlations in winter, which is similar with what we found for the Rhine basin. For every season, correlations are lower with the observations compared to the simulations, especially in MAM. As this is the cropping period, irrigation requires a lot of
10 water and reduces substantially the observed streamflow. Irrigation is currently not included in the hydrological model. This result shows the importance of including human activities in hydrological models.

The selected event over the Mississippi basin (open circle Fig. 15b) occurs in January and corresponds to both an annual maxima in the 10-day precipitation sum (66.8 mm) as well to an annual maxima in discharge (72 000 m$^3$ s$^{-1}$). In addition, the selected event is the second most extreme event in the DJF Gumbel plot for precipitation (Fig. 12a). From the synoptic situation
and the vertical integrated moisture fluxes in Fig. 16b we conclude that moisture is mainly transported from the Pacific and the Gulf of Mexico leading to precipitation over the South-East of the Mississippi basin, which is a region prone for extreme





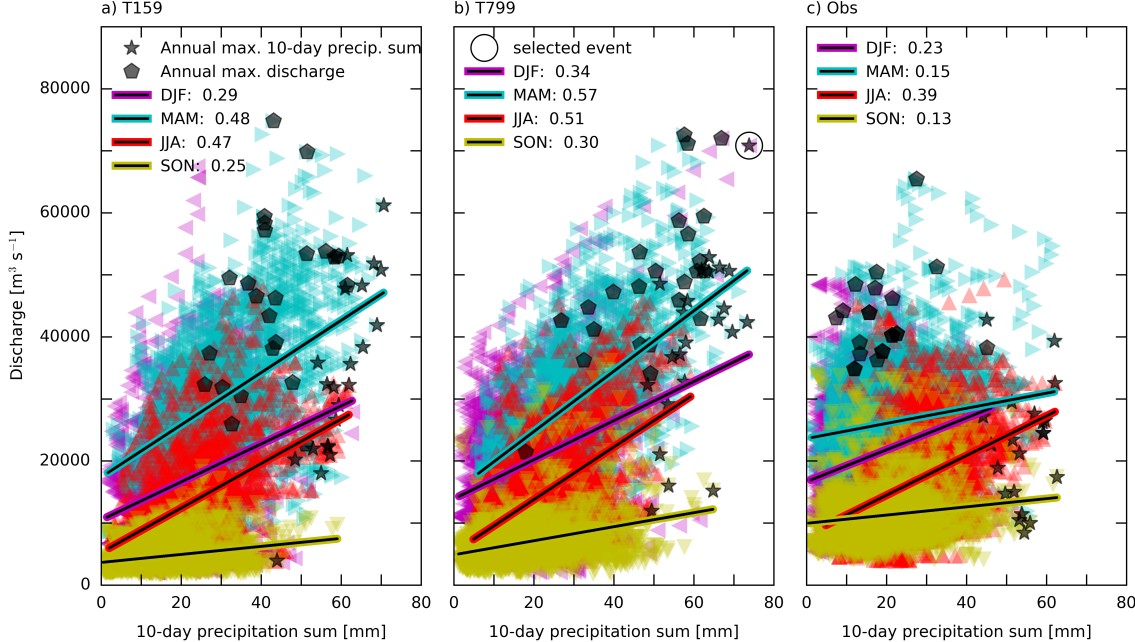

**Figure 15.** Scatterplot for the Mississippi basin of daily discharge [m$^3$ s$^{-1}$] with previous 10-day precipitation sums [mm] for a) the low resolution forcing (T159), b) the high resolution forcing (T799) and c) the observations (Obs). The discharge results shown here are obtained with the 0.5° GHM. The different seasons are indicated with the colours and regression line and correlation value. The annual maxima of both 10-day precipitation sums and discharge are indicated with respectively the black stars and hexagons.

precipitation (Wehner et al., 2010). Berghuijs et al. (2016) shows that (multiple) large precipitation events mainly occur over the South-East of the USA during winter. As the precipitation falls very close to Vicksburg, the response in discharge is relatively quick and leads to an exceptional high discharge (72 000 m$^3$ s$^{-1}$), with a return period of 30 years.

# 5  General discussion

## 5.1  Discussion on methodology

In this study, we force a GHM with a GCM (as illustrated in Fig. 3) to simulate the hydrological cycle for two large rivers basins. To simulate the land-surface hydrology, we have chosen to use a GHM rather than the GCMs land-surface model (LSM), as most GCMs (including EC-Earth) do not have a detailed routing module. In addition, actual evaporation is calculated on a higher resolution in a GHM compared to a LSM.

To quantify the impact of resolution in the atmosphere and in hydrology, we performed a systematic comparison of high and low resolutions simulations (Fig. 1). Such computations are time consuming and computationally expensive, especially when we force the high-resolution GHM with the high-resolution GCM. Therefore, this study is limited to two selected basins and



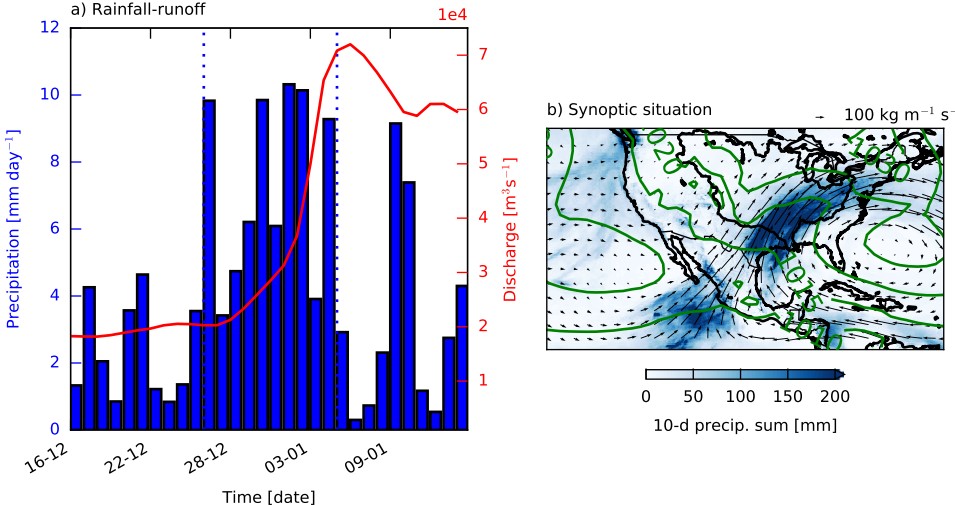

**Figure 16.** In a) precipitation (in blue) and discharge (in red) for the Mississippi are shown 20 days before and 10 days after the selected event. The vertical dotted lines indicate the 10 day period, which is spatially summed in b). The contour lines in b) indicate the 10-day average mean sea level pressure in hPa and the arrows the 10-day averaged vertical integrated moisture fluxes in kg m$^{-1}$ s$^{-1}$.

one particular climate- and hydrological model. Currently, there are only few global climate model simulations available at such high resolution as EC-Earth T799 (Davini et al. 2017b; Delworth et al. 2012; Schiemann et al. 2018; and more references within Haarsma et al. 2016). An advantage of the EC-Earth experiments is that we can compare different resolutions within the same set of physical parameterizations. A limitation is that the resolution of EC-Earth T799 is still not high enough to capture

all relevant local hydrological processes related to local orography and convection. The study by Schiemann et al. (2018) used a similar set-up of increased model resolution (towards ∼ 25 km with the HadGEM model) but related the improvements in winter precipitation over Europe mainly to a better resolved orography while they concluded that resolution sensitivity to the storm-track plays a lesser role. This illustrates that the sensitivity of the model resolution is also depending on the GCM. In the HighResMIP project, the robustness of climate simulations at high resolution (∼ 25 km) will be examined (Haarsma

et al., 2016). Further (high-resolution) model comparison in conjunction with (high-resolution) hydrological modelling will be needed for a global diversity of catchments.

    Within this resolution study, we have only performed a limited performance analysis of W3RA. The study by Beck et al. (2017) compares daily runoff from multiple GHMs and found pronounced inter-model performance differences. This underscores the importance of hydrological model uncertainty. The GHM W3RA obtained moderate to good scores in the study

by Beck et al. (2017). Our results show that W3RA overestimates actual evaporation which results in an underestimation of discharge in both the Rhine and the Mississippi basin.

    An increase in horizontal resolution results in an improved representation of the landscape, i.e. improved spatial heterogeneities in topography, soil and vegetation. This is beneficial for simulating hydrometeorological processes for both the GCM





and the GHM. In case of an increased resolution GCM, large-scale dynamics will be better represented, while mesoscale phenomena and moist and radiative processes are still parameterized. On the other hand, in GHMs, most relevant processess are parameterized. With increasing resolutions, processes like lateral groundwaterflow become more important and need to be resolved explicitly (Wood et al., 2011; Bierkens et al., 2015; Van Dijk, 2010b). Furthermore, the parameters itself are also un-

certain, especially because most are non-physical and difficult to determine across scales (Melsen et al., 2016). Several studies apply parameter regionalization techniques to adapt parameters across scales (Samaniego et al., 2010). To allow for a fair comparison between the different resolutions GHM, we simply remapped the parameters from the low to the high resolution model, except for vegetation and orography which are known at high resolution (see Sect. 3.2). With this technique, it is possible to see the improvements in monthly averaged precipitation, due to the higher resolution GCM, reflected in the monthly averaged

discharge results. We find that maximum and minimum discharge are not consistently improved with higher resolution GCM and GHM. This could be related to the choice of our GHM, as we only use one model, the fact that the key processes to model these extremes are not well represented in the model, or that the input data ($\sim 25$ km) is still not sufficient to capture all the processes to simulate the hydrological extremes.

## 5.2    Broader implications

This study focuses on the effect of resolution of a GCM and a GHM in simulating the hydrological cycle for the Rhine and Mississippi basins. These two well measured basins only represent a subsample of the global diversity of catchments. Nonetheless, the conclusions from this study could be used as a guideline when assessing the benefits of resolution increases in modelling the hydrological cycle of other basins with comparable characteristics. The Rhine basin is mainly driven by large-scale dynamics, and the improvements in precipitation are also attributed to an improved representation of these large-scale

processes (i.e. improved storm-track; Van Haren et al. (2015)). Our conjecture is that, for basins situated along the mid-latitude storm-track, improvements in the hydrological cycle can be obtained when high resolution global climate models are used. For the Mississippi basin, no clear improvements in precipitation were found with increased resolution. This can most likely be explained by EC-Earth being a hydrostatic model. Therefore, when studying the hydrological cycle over basins of which the precipitation is mostly driven by convection, we recommend to use convection permitting models, to more explicitly resolve

tropical storms and moist convective processes (Liu et al., 2017; Prein et al., 2017).

## 6    Summary and conclusions

We study the benefits of increased spatial resolution from both the climate and hydrological perspective by forcing a global hydrological model with a global climate model. We analyse three main components of the hydrological cycle: precipitation, actual evaporation and discharge. We focus on two river basins with contrasting climatic drivers and for which enough

verification data exists: the Rhine and Mississippi basins.

By increasing the resolution of the EC-Earth GCM from $\sim 120$ km$^2$ to $\sim 25$ km$^2$, precipitation over the Rhine basin improves significantly, caused by the better represented large-scale circulation patterns (Van Haren et al., 2015). Therefore, we



suggest to use high resolution simulations on a global scale when studying climatic impacts on the Rhine basin. The climatic drivers of the Mississippi basin are local convective events, large-scale systems from the Pacific and moisture transport from the Caribbean, possibly associated with tropical storms. Our results show that the increased resolution GCM ($\sim 25$ km$^2$) hardly affects precipitation over the Mississippi basin. Likely this is because of the dependence of precipitation on local parametized

physical processes over the Mississippi region. These convective parameterizations are similar for both resolutions. For a good representation of the hydrological cycle over the Mississippi basin, we therefore recommend to use convection resolving models to explicitly resolve moist convective processes.

To increase the model resolution of the GHM, we have remapped the parameters from the 0.5° to the 0.05° resolution, except for orography and vegetation. With these settings for the high resolution GHM ($\sim 5$ km$^2$), no significant changes in discharge

were found for both basins. Improvements in discharge are expected with high resolution GHMs when hydrological processes and parameters are better understood and described. Nevertheless, (improved) monthly averaged precipitation from the GCM is reflected in (improved) monthly averaged actual evaporation and discharge from the GHM. Thus, the monthly averaged hydrological cycle of the Rhine is better simulated with high-resolution GCM input, although we did not found improvements in the representation of extreme streamflow events. For the Mississippi basin, no substantial differences in precipitation and

discharge were found between the two resolutions input GCM and the two resolutions GHM. Based on the results of our study, we conclude that due to the clearly distinct response of the chosen river basins to resolution increase, the route from improved resolution to better results is a challenging one. Our study, however, provides new and valueable insights on what to expect when modelling the hydrological cycle for basins in the midlatitude storm tracks or in convection-dominated regions.

## 7  Code and data availability

The observational data (precipitation from E-OBS and CPC, actual evaporation from GLEAM and discharge from GRDC) used in this study are stored in a repository: https://doi.org/10.4121/uuid:b7b988fc-f5c8-4ce1-8e33-47f31d04a99d. The parameter fields of the hydrological model and the routing module are also stored in this repository, together with the executables of these models. The EC-Earth data and the output of the hydrological model is avaiable upon request to the authors.

*Competing interests.*  There are no competing interests.



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
