# Peer review of "The benefits of spatial resolution increase in global simulations of the hydrological cycle evaluated for the Rhine and Mississippi basins"

_Hydrology and Earth System Sciences, 2018_

## Referee Comment (RC1) · M.-E. Demory (Referee) · 16 Oct 2018

I would like to acknowledge the authors' hard work in revising their manuscript and addressing all of my concerns. The single case events are now better put into context, the conclusions focused on the two studied basins are better justified, and the moisture fluxes strengthen their conclusions as expected. It is also a good idea to add a broader implications section. I find the manuscript much improved and now suitable for publication.

Kind regards

---

## Referee Comment (RC2) · HHG Savenije (Referee) · 18 Oct 2018

The authors study the potential improvement of rainfall-runoff prediction by improving the resolution of both a climate/weather model and a hydrological model. In essence, this is a study where the hypothesis is that smaller resolution leads to better representation. This is a reductionist vision, which conflicts with the view that systems need to be described at the correct scale at which the dominant physical processes generate runoff. The question should be whether a reduction of scale still contributes to better process description. The correct scale depends on the landscape. Different landscape types (defined by topographical and ecosystem characteristics) house different runoff

mechanisms, such as there are: saturation excess overland flow, saturation excess subsurface flow, Hortonian overland flow and fast and slow groundwater recession. These processes are physical and if connected to the correct landscape indicators can be regionalised without further calibration (e.g. Gao et al., 2016; 2014a). So instead of merely reducing the resolution of a (lumped) conceptual model, it is better to account for landscape in more detail, while retaining parameterization based on the dominant runoff mechanisms pertaining to these landscapes. The authors recognise as much on page 25, line 17, and in the following paragraph on page 26, but they don't seem to take it into practice.

The weakness of the paper is that there is no insight whatsoever in the hydrological model used. The references to Van Dijk 2010a,b are insufficient. This is grey literature about a model developed for Australia, which I cannot find on the web and which gives the reader no insight into the working of this model. There is also no indication whether this model (calibrated for Australian circumstances, I presume) would work for the Mississippi or the Rhine as well. Probably, it is a semi-distributed conceptual model that distinguishes between deep and shallow-rooting vegetation to represent the ecosystems. This may be a good and appropriate model for Australia, but without a description of the main characteristics of the model, the effect of scale in the hydrological response cannot be evaluated.

The most crucial parameter in any hydrological model is the storage capacity that the vegetation has created in the root zone. Surprisingly, this root zone storage capacity is independent on the soil parameters, because the vegetation adjusts the rooting depth to the soil, so as to create sufficient buffer capacity to overcome drought. This root zone storage capacity is scale-independent and directly connected to climate (Gao et al., 2014b), can be applied globally (Wang-Erlandsson et al., 2016) and locally (Boer-Euser et al., 2016), outperforming traditional soil-based approaches. So instead of using soil information and rooting depth as the main (and highly unreliable) input to hydrological models, it would be much better –and much easier – to use the scaleindependent climate-based root zone storage capacity as the key input. I am curious to hear the opinion of the authors on this issue.

My assessment:

I do consider this a well-written and well-prepared paper, but I find the research question not very innovative: testing with an ill-described model (probably developed for local circumstances) whether a reduction of resolution results in better performance. Runoff processes are largely determined by climate, ecosystem and topography. A model that requires calibration of scale-dependent parameters is not suitable for such an exercise. Although I support the conclusion about the reduction of scale in the meteorology and the difference between the dominant rainfall bringing mechanisms in the Mississippi and the Rhine, I doubt the adequacy of the study on reducing the scale of the hydrological model. The authors apparently missed a considerable part of the debate on hydrological modelling, as for instance presented in Hrachowitz et al. (2013), where these issues were summarized after an intensive debate among the entire hydrological community.

References:

Boer-Euser, T., H. K. McMillan, M. Hrachowitz, H. C. Winsemius, and H. H. G. Savenije, 2016. Influence of soil and climate on root zone storage capacity, Water Resour. Res., 52, 2009–2024, doi:10.1002/2015WR018115.

Gao, H., M. Hrachowitz, F. Fenicia, S. Gharari, and H. H. G. Savenije, 2014a. Testing the realism of a topography driven model (FLEX-Topo) in the nested catchments of the Upper Heihe, China, Hydrol. Earth Syst. Sci., 18, 1895-1915, 2014a.

Gao, H., M. Hrachowitz, S.J. Schymanski, F. Fenicia, N. Sriwongsitanon, H.H.G. Savenije, 2014b. Climate controls how ecosystems size the root zone storage capacity at catchment scale, Geophysical Research Letters, 41, 7916-7923, doi: 10.1002/2014GL061668.

Gao, H., M. Hrachowitz, N. Sriwongsitanon, F. Fenicia, S. Gharari, and H. H. G. Savenije, 2016. Accounting for the inïñĆuence of vegetation and landscape improves model transferability in a tropical savannah region, Water Resour. Res., 52, 7999–8022, doi:10.1002/2016WR019574.

Hrachowitz, M., Savenije, H.H.G., Blöschl, G., McDonnell, J.J., Sivapalan, M., Pomeroy, J.W., Arheimer, B., Blume, T., Clark, M.P., Ehret, U., Fenicia, F., Freer, J.E., Gelfan, A., Gupta, H.V., Hughes, D.A., Hut, R.W., Montanari, A., Pande, S., Tetzlaff, D., Troch, P.A., Uhlenbrook, S., Wagener, T., Winsemius, H.C., Woods, R.A., Zehe, E., and Cudennec, C., 2013. A decade of Predictions in Ungauged Basins (PUB); a review. Hydrological Sciences Journal, 58 (6), 1–58, doi: 10.1080/02626667.2013.803183.

Wang-Erlandsson, L., W. G. M. Bastiaanssen, H. Gao, J. Jägermeyr, G. B. Senay, A. I. J. M. van Dijk, J. P. Guerschman, P. W. Keys, L. J. Gordon, and H. H. G. Savenije, 2016. Global root zone storage capacity from satellite-based evaporation. Hydrol. Earth Syst. Sci., 20, 1459–1481, doi:10.5194/hess-20-1459-2016.

---

## Referee Comment (RC3) · Anonymous Referee #3 · 24 Oct 2018

I thank the authors for this revised version and for addressing my comments. In this revised version, the authors recognise that an increase in resolution does not necessarily lead to better streamflow simulations, and clearly state it in the abstract and in the main text. This made me wonder why it was not the case, so I went through the different versions of the manuscript, reviewers' comments and authors' replies. In a nutshell, I am concerned that i) the GHM was not ideally set up and ii) the experimental setup chosen by the authors prevented them from fully exploiting the benefits of the higher resolution.

GHM setup: When forced using ERAI, the GHM severely underestimates streamflow

in the Rhine basin (Figure 5c), in particular in summer, when as much as half of the streamflow is missing. The authors recognise that the "GHM is too dry in the summer months", since ERAI precipitation estimates are "good". I raised this issue in a previous round of revisions, and the authors replied that they "have not performed an in-depth analysis on the performance of W3RA as this study focuses on the sensitivity to resolution." I regret to say that I do not agree with this argument. Before increasing the resolution, the authors should have made sure that the basic hydrological behaviour of the basin (its water balance) is adequately captured. In this specific case, the streamflow underestimation in summer indicates that the GHM is not adequately setup for the Rhine catchment. From a more general perspective, the authors mentioned in their reply that "The global hydrological model which we use is not calibrated for the specific catchments of this study. In general, calibration of global hydrological models is limited." I recognise that, when run at the global scale, GHMs are challenging to calibrate, but for this study, I consider that the authors should have taken the time to adjust model parameters in order to provide acceptable streamflow simulations for the two basins they selected, before increasing the GHM resolution.

Parameter estimation under higher resolution: In the revised manuscript, the authors mention that for the GHM, "to allow a fair comparison between the two model resolutions, we remapped these parameters from the 0.5° to the 0.05° resolution." The authors explained in a previous round of reviews that "we remap the parameters from the low to the high resolution using the resample statement in PCRaster (Karssenberg et al., 2010)". How the remapping exactly works is unfortunately not documented in the revised manuscript, but my impression is that a resampling routine alone is not sufficient to incorporate the new data and knowledge necessary to enable the model to resolve smaller-scale processes. In their revised abstract, the authors report that "Increasing the resolution of vegetation and orography in the high resolution GHM (from 0.5° to 0.05°) shows no significant differences in discharge for both basins, likely because the hydrological processes depend highly on model parameter values that are not readily available at high resolution." All this indicates that although the GHM was
run at higher resolution, it was probably run using parameters at an effective resolution close to the original (coarser) resolution, because model parameters are higher resolution were not "readily available". This likely explains why the benefits of increasing the GHM resolution were limited. Arguably, increasing the resolution of a model goes beyond decreasing its grid spacing, it should also involve the incorporation of new data and knowledge on process representation across scales. It is my impression that this part is essentially missing and this is preventing us from truly assessing the benefits of the increased resolution.

In conclusion, this study addresses an interesting research question, and it is clear that a lot of effort has gone into it. However, I consider that the experimental setup presents fundamental flaws, which cannot be easily corrected (it would be necessary to re-run most of the analysis to fix them), and which significantly limit the insights the authors (and the community) can gain into the benefits of model resolution increase.

---

## Editor Comment (EC1) · S. J. Schymanski (Editor) · 25 Oct 2018

In RC2 and RC3, the referees touch on an important question: is it meaningful to increase the resolution of a hydrological model if we do not know whether it represents the relevant processes adequately? It is unfortunate, but at the same time revealing that this question only arose after the recognition that increasing model resolution did not lead to improved streamflow simulations (RC3). At the same time, there is strong advocacy in the scientific community to use growing computational capabilities to increase model resolution. From this perspective, this study could be seen as a reminder that increasing model resolution is not meaningful if the model does not represent the

governing processes at the target resolution. The key question is how to test if a model does represent the relevant processes adequately. Both referees point out that an increase in hydrological model resolution likely requires changes in paramterizations and/or process representations. I wonder how many other studies have neglected this fact and what are the definitive lessons that can be learnt from the present study.

———————————————————

---

## Author Comment (AC1) · 5 Nov 2018

We would like to thank Marie-Estelle for the useful comments throughout the review process which substantially improved the manuscript. We are happy to hear that she recognizes our efforts to revise the manuscript and that she is satisfied with the result.

Best regards, On behalf of all co-authors,

Imme Benedict

---

## Author Comment (AC2) · 5 Nov 2018

*We would like to thank prof. Huub Savenije for his comments on our manuscript. We reply to the raised points below in italic.*

The weakness of the paper is that there is no insight whatsoever in the hydrological model used. The references to Van Dijk 2010a,b are insufficient. This is grey literature about a model developed for Australia, which I cannot find on the web and which gives the reader no insight into the working of this model. There is also no indication whether this model (calibrated for Australian circumstances, I presume) would work for the Mississippi or the Rhine as well. Probably, it is a semi-distributed conceptual model that distinguishes between deep and shallow-rooting vegetation to represent the ecosystems. This may be a good and appropriate model for Australia, but without a description of the main characteristics of the model, the effect of scale in the hydrological response cannot be evaluated.

*We will add more peer reviewed references of the W3RA model to the manuscript (Van Dijk, 2010b; Van Dijk and Renzullo, 2011; Van Dijk et al., 2012a). The main evaluation of the model on global scale is documented in Van Dijk (2013) in a Water Resources paper. The main description of the model code and parameters is given here (Van Dijk, 2010):*
http://www.clw.csiro.au/publications/waterforahealthycountry/2010/wfhc-aus-water-resources-assessment-system.pdf
*The model code is also open source and online available on Github:*
https://github.com/openstreams/wflow/blob/master/wflow-py/wflow/wflow_w3ra.py

*W3RA is a global model and therefore not specifically calibrated/tested for the Rhine and Mississippi regions. The model is indeed a semi-distributed conceptual model and it distinguishes between deep and shallow-rooting vegetation to represent the ecosystems (as mentioned in line 5 on page 5 in the manuscript). For an extensive description we refer to the documentation of the model, which presents all equations and parameters (see document above; Van Dijk, 2010). Furthermore, we would like to indicate more studies which use W3RA for analysing discharge at catchment scale (van Dijk, 2014; Beck et al., 2017; Schellekens et al 2017; van Dijk 2013), from which van Dijk (2014) and Beck et al. (2017) are published in HESS. We will edit the manuscript to ensure that the full description of the model and its source code is retrievable from the cited references.*

The most crucial parameter in any hydrological model is the storage capacity that the vegetation has created in the root zone. Surprisingly, this root zone storage capacity is independent on the soil parameters, because the vegetation adjusts the rooting depth to the soil, so as to create sufficient buffer capacity to overcome drought. This root zone storage capacity is scale-independent and directly connected to climate (Gao et al., 2014b), can be applied globally (Wang-Erlandsson et al., 2016) and locally (Boer-Euser et al., 2016), outperforming traditional soil-based approaches. So instead of using soil information and rooting depth as the main (and highly unreliable) input to hydrological models, it would be much better –and much easier – to use the scale- independent climate-based root zone storage capacity as the key input. I am curious to hear the opinion of the authors on this issue.

*We thank the reviewer for his comment. We agree that the rooting depth and root-zone storage (and crop factors) play an important role in hydrological models. We agree that this novel approach deserves full attention in the global hydrological modelling and therefore should be studied further. Application and testing of approaches developed by Wang-Erlandsson et al (2016) and others, or going to even more dynamical root density approaches (e.g. van Wijk and Bouten, 2001), is very interesting, in particular because of the scale independence . When reading the comments we realize we have not given full attention to these innovating topics on hydrological modelling and we will elaborate more on them in the discussion and suggest further work on it.*

My assessment:
I do consider this a well-written and well-prepared paper, but I find the research question not very innovative: testing with an ill-described model (probably developed for local circumstances) whether a reduction of resolution results in better performance. Runoff processes are largely determined by climate, ecosystem and topography. A model that requires calibration of scale-dependent parameters is not suitable for such an exercise. Although I support the conclusion about the reduction of scale in the meteorology and the difference between the dominant rainfall bringing mechanisms in the Mississippi and the Rhine, I doubt the adequacy of the study on reducing the scale of the hydrological model. The authors apparently missed a considerable part of the debate on hydrological modelling, as

for instance presented in Hrachowitz et al. (2013), where these issues were summarized after an intensive debate among the entire hydrological community.

*We thank prof. Savenije for the assessment. We are aware of the ongoing debate about scales and hydrological modelling, and we have sharpened our knowledge on root-zone storage capacity. We would like to emphasize that we do not claim that the approach taken here is the best. We have tested from a 'global modelling perspective' if scaling up the resolution will lead to better performance (going from 0.5 to 0.05 degrees). We have investigated if rescaled parameters, and only using high-resolution information we are certain about (topography and vegetation), at higher resolution would give better simulation results. This also allows comparing the outcome of the models in a fair and transparent way, which would not be possible or be very difficult otherwise, because the model itself would change. As concluded in the manuscript, we find that the improvements are limited and that likely other process representation (e.g. subsurface lateral flow) is needed, especially if we move to even finer resolution than 0.05 degrees.*

*The comments of Huub Savenije stress the challenges in the field of hydrological modelling, especially at different scales. We hope this manuscript can contribute to this discussion by showing the results of one approach to deal with different spatial scales, which we conclude is not the best method. We will include an extra paragraph in the discussion to put our approach in perspective with the challenges in hydrological modelling across scales, and the potential benefits of a different viewpoint, illustrated along the suggested references by the reviewer.*

References

Beck, H. E., van Dijk, A. I. J. M., de Roo, A., Dutra, E., Fink, G., Orth, R., and Schellekens, J.: Global evaluation of runoff from 10 state-of-the-art hydrological models, Hydrol. Earth Syst. Sci., 21, 2881-2903, https://doi.org/10.5194/hess-21-2881-2017, 2017.

Schellekens, J., Dutra, E., Martínez-de la Torre, A., Balsamo, G., van Dijk, A., Weiland, F.S., Minvielle, M., Calvet, J.C., Decharme, B., Eisner, S. and Fink, G., 2017. A global water resources ensemble of hydrological models: the eartH2Observe Tier-1 dataset. *Earth System Science Data*, *9*(2), pp.389-413.

Van Dijk, A. I. J. M. (2010b), AWRA Technical Report 3, Landscape Model (Version 0.5) Technical Description, WIRADA/CSIRO Water for a Healthy Country Flagship, Canberra. [Available at http://www.clw.csiro.au/publications/waterforahealthycountry/2010/wfhc-aus-water-resources-assessment-system.pdf ]

Van Dijk, A. I. J. M., and L. A. Bruijnzeel (2001), Modelling rainfall interception by vegetation of variable density using an adapted analytical model. Part 1: Model description, J. Hydrol., 247(3–4), 230–238.

Van Dijk, A. I. J. M., and L. J. Renzullo (2011), Water resource monitoring systems and the role of satellite observations, Hydrol. Earth Syst. Sci., 15, 39–55.

van Dijk, A. I. J. M., Renzullo, L. J., Wada, Y., and Tregoning, P.: A global water cycle reanalysis (2003–2012) merging satellite gravimetry and altimetry observations with a hydrological multi-model ensemble, Hydrol. Earth Syst. Sci., 18, 2955-2973, https://doi.org/10.5194/hess-18-2955-2014, 2014.

van Dijk, A.I., Peña-Arancibia, J.L., Wood, E.F., Sheffield, J. and Beck, H.E., 2013. Global analysis of seasonal streamflow predictability using an ensemble prediction system and observations from 6192 small catchments worldwide. *Water Resources Research*, *49*(5), pp.2729-2746.

Van Wijk, M.. and W Bouten, 2001. Towards understanding tree root profiles: simulating hydrologically optimal strategies for root distribution, HESS, 629-644, doi: 10.5194/hess-5-629-2001.

---

## Author Comment (AC3) · 5 Nov 2018

*We thank reviewer 3 for the constructive comments. Below we give the detailed comments to most of his/her points in italic.*

GHM setup: When forced using ERAI, the GHM severely underestimates streamflow in the Rhine basin (Figure 5c), in particular in summer, when as much as half of the streamflow is missing. The authors recognise that the "GHM is too dry in the summer months", since ERAI precipitation estimates are "good". I raised this issue in a previous round of revisions, and the authors replied that they "have not performed an in-depth analysis on the performance of W3RA as this study focuses on the sensitivity to resolution." I regret to say that I do not agree with this argument. Before increasing the resolution, the authors should have made sure that the basic hydrological behaviour of the basin (its water balance) is adequately captured. In this specific case, the streamflow underestimation in summer indicates that the GHM is not adequately setup for the Rhine catchment.

*In our analysis, we found comparable simulated discharge with ERAI for the Rhine as Photiadou et al (2011), who used a high resolution HBV96 calibrated hydrological model for the Rhine forced with ERAI. In contrast to their claims of underestimated precipitation, we found that the total precipitation over the Rhine catchment in ERAI is comparable with the observations. Due to resolution issues the distribution of the rainfall over the catchment might not be optimal, which could be important for snow build up in the Alps and hence could cause too low discharges in spring/summer. We found that the total actual evaporation over the Rhine basin is overestimated when compared to GLEAM. This could further contribute to the simulation results we obtained. However, the observed precipitation, and in particular evaporation contain large uncertainties (Bosilovich, 2008 ; Miralles, 2011). In our manuscript, we present a transparent and fair assessment of the mismatch between model and data. We could conduct a more in depth analysis of this issue. However, in our opinion, this will hardly contribute to the main aim of this work, which is testing the effect of resolution on simulating the hydrological cycle. We leave it up to the editor to decide if additional analysis is needed.*

From a more general perspective, the authors mentioned in their reply that "The global hydrological model which we use is not calibrated for the specific catchments of this study. In general, calibration of global hydrological models is limited." I recognise that, when run at the global scale, GHMs are challenging to calibrate, but for this study, I consider that the authors should have taken the time to adjust model parameters in order to provide acceptable streamflow simulations for the two basins they selected, before increasing the GHM resolution.

*This study focuses on the effect of increased resolution in simulating the hydrological cycle of river basins. Therefore, we use a global climate model and a global hydrological model. We examine our hypothesis by focussing on two large river basins, with different characteristics. To decrease the computing time we only run the global hydrological model for the two selected basins. However, we still do this from the viewpoint of 'global hydrology', i.e. using a global hydrological model not specifically designed for the two basins. If we want to obtain the best results in modelling the hydrological basins over the Rhine and Mississippi basin, we should have chosen specific regional models build for the specific basins, which is not the research question of the study. We intend to add an extra paragraph on the issue on global modelling and calibrated catchment scale modelling in response to reviewer 3.*

Parameter estimation under higher resolution: In the revised manuscript, the authors mention that for the GHM, "to allow a fair comparison between the two model resolutions, we remapped these parameters from the 0.5_ to the 0.05_ resolution." The authors explained in a previous round of reviews that "we remap the parameters from the low to the high resolution using the resample statement in PCRaster (Karssenberg et al., 2010)". How the remapping exactly works is unfortunately not documented in the revised manuscript, but my impression is that a resampling routine alone is not sufficient to incorporate the new data and knowledge necessary to enable the model to resolve smaller-scale processes. In their revised abstract, the authors report that "Increasing the resolution of vegetation and orography in the high resolution GHM (from 0.5_ to 0.05_) shows no significant differences in discharge for both basins, likely because the hydrological processes depend highly on model parameter values that are not readily available at high resolution." All this indicates that although the GHM was run at higher resolution, it was probably run using parameters at an effective resolution close to the original (coarser) resolution, because model parameters are higher resolution were not "readily available". This likely explains why the benefits of increasing the GHM resolution were limited. Arguably, increasing the resolution of a model goes beyond decreasing its grid spacing, it

should also involve the incorporation of new data and knowledge on process representation across scales. It is my impression that this part is essentially missing and this is preventing us from truly assessing the benefits of the increased resolution.

*In every version of the manuscript we mentioned "to allow a fair comparison between the two model resolutions, we remapped these parameters from the 0.5 to the 0.05 degrees resolution." Apparently, this was not clear enough and we will adjust the text to make this procedure clearer.*
*The reviewer understood correctly that in this study we only tested the impact of increased resolution by increasing resolution and not by including more/different smaller-scale processes or involving new data. We decided to do so, since applying all suggested adaptions to the high-resolution version would imply a different model which is not directly comparable with the low resolution version. Therefore, we investigated if rescaling parameters, and only using known high-resolution parameters (topography and vegetation), at higher resolution would give better results. It was shown by Melsen et al (2016) that parameters can be transferred across the spatial scales, on regional scales from 1 $km^2$ to 100 $km^2$, and our work could be seen as a large-scale test of their work. We conclude our paper by stating exactly the view of the reviewer: quick gains are not straightforwardly achieved (line number 9-10-11 on page 27).*

References

Bosilovich, M.G., Chen, J., Robertson, F.R. and Adler, R.F., 2008. Evaluation of global precipitation in reanalyses. *Journal of applied meteorology and climatology*, *47*(9), pp.2279-2299.

Melsen, L., Teuling, A., Torfs, P., Zappa, M., Mizukami, N., Clark, M., and Uijlenhoet, R.: Representation of spatial and temporal variability in large-domain hydrological models: case study for a mesoscale pre-Alpine basin, Hydrol. Earth Syst. Sci., 20, 2207-2226, https://doi.org/10.5194/hess-20-2207-2016, 2016.

Miralles, D.G., De Jeu, R.A.M., Gash, J.H.C., Holmes, T.R.H. and Dolman, A.J., 2011. Magnitude and variability of land evaporation and its components at the global scale.

Photiadou, C. S.,Weerts, A. H., and van den Hurk, B. J. J. M.: Evaluation of two precipitation data sets for the Rhine River using streamflow simulations, Hydrology and Earth System Sciences, 15, 3355–3366, doi:10.5194/hess-15-3355-2011, 2011.

---

## Author Comment (AC4) · 5 Nov 2018

We would like to thank the editor dr. Schymanski for summarizing the main outcome of RC2 and RC3 and we endorse his assessment that this paper can be seen as a reminder that increasing model resolution alone is not meaningful if the model does not represent the governing processes at the target resolution. Indeed, an important question is if models represent processes adequately and what we mean with 'adequately'. This question is almost always context-dependent. Relative few studies focus on new/better process representation but rather focus more on calibration/ model parameters when changing scales (see also Melsen et al 2016 and references therein).

[Figure]

The discussion triggered in this round of reviews stresses the still challenging field of hydrological modelling at multiple scales. The scale interaction from small to large scales and vice versa is insufficiently understood and different approaches deserve rigorous study. Here we followed one approach. We hope this manuscript can contribute to this discussion by showing the results of this, potentially not best, approach to deal with multiple scales and scale interactions. We will elaborate the discussion in order to put our approach in perspective with the challenges in hydrological modelling, and we will add the references suggested by the reviewer.

Best regards, On behalf of all co-authors, Imme Benedict

---

## Author Response (AR1)

Dear authors,

Thank you for your open and informative responses to the referee comments and to my own. I believe that the paper will be a valuable contribution to the scientific literature, once the changes you suggested have been implemented.

I do not believe that additional analysis is necessary to make the main points of your study, i.e. that increasing resolution of a GHM does not necessarily improve hydrological prediction. The referees' critique that the chosen GHM may not adequately represent hydrological processes at the chosen catchments would likely apply to most other GHMs as well, while tests of adequate representation of processes are not easily performed. Your case study may not be able to answer the general question if increased GHM resolution has the potential to improve predictions, but it does contribute some evidence for the contrary conclusion. This raises the question whether we should not invest in improving process representation instead of increasing spatial resolution. The points raised by the referees and your replies are valuable pieces of this discourse and I look forward seeing them reflected in your revised paper.

Could you also provide a script or instructions on how to reproduce the results using the data mentioned in "Code and data availability"? I looked at the link but got the impression that it may not be straightforward to reproduce your results with the information given.

Dear dr. Schymanski,

We like to thank you for your clear summary of the reviewers' comments, your comment and our replies.

Following the line of the open discussion, we have adapted the manuscript as follows:
- We have re-structured the discussion section. We start the discussion by indicating that this study is constrained by the use of one GCM, one GHM and two river basins. Following these constrains we have made the following sections in the discussion: 5.1 Increased resolution of the global climate model, 5.2 Process presentation vs. increased resolution of the global hydrological model, and 5.3 Extrapolation of results to other basins
  - The first section 5.1 Increased resolution of the global climate model, contains similar information as was already present in the previous version of the discussion.
  - 1.2 Process representation vs. increased resolution of the GHM: In this section we put our approach into the larger perspective on the discussion of process representation and increased resolution.
    We added the references suggested by the reviewers, for example using root-zone storage capacity as scale-independent parameter. We tried to emphasize that our approach of simply increasing resolution is one way to go but as the results indicate probably not the best approach. Therefore we recommend to invest in better process presentation (see section 1.2)
    At the end of the section we also added a paragraph to emphasize that all simulations are done from the viewpoint of 'global hydrology', i.e. using a global hydrological model not specifically designed for the two basins. And that if we wanted to obtain the best modelling results of discharge for the two basins we should have used calibrated-catchment models, but that this is not the purpose of this study. We also mention that GHMs are often used in climate context, while not begin optimized for individual basins.
  - We renamed section 5.3 from 'Broader implications' to 'Extrapolation of results to other basins', to better indicate the content of this section.

- We have adapted the abstract and summary to clarify our approach/viewpoint and methodology. Furthermore, we have slightly changed the order in which we present the results in this two sections.
- We have added extra references on the W3RA model in the method section and a link to the code on Github in the method and data availability section.
- We have added the study from Van der Wiel et al., 2018 on modelling floods in the Mississippi basin, this paper recently appeared in Journal of Hydrometeorology.
- We corrected some small mistakes throughout the manuscript
- We have changed the title to better indicate our approach

We included a diff file where all exact changes made to the manuscript are indicated. We included the responses to the reviewers as presented in the open discussion and we have added in blue what we have adapted to the manuscript based on their comments.

As a response to your last point on code and data availability, we have added an extra document (README) to the repository to explain how to create time-series from the spatial fields of evaporation and precipitation given in the repository (https://data.4tu.nl/repository/uuid:c3b6e367-8215-4640-81d2-9f74994e65f4). We refer to this document in the code and data availability section.

With kind regards,

Imme Benedict, on behalf of all co-authors

**Review Huub Savenije**

*We would like to thank prof. Huub Savenije for his comments on our manuscript. We reply to the raised points below in italic, these are the replies from the open discussion. The replies in blue are additional to the open discussion replies, and indicate the changes that we have made in the manuscript.*

Reviewer: The weakness of the paper is that there is no insight whatsoever in the hydrological model used. The references to Van Dijk 2010a,b are insufficient. This is grey literature about a model developed for Australia, which I cannot find on the web and which gives the reader no insight into the working of this model. There is also no indication whether this model (calibrated for Australian circumstances, I presume) would work for the Mississippi or the Rhine as well. Probably, it is a semi-distributed conceptual model that distinguishes between deep and shallow-rooting vegetation to represent the ecosystems. This may be a good and appropriate model for Australia, but without a description of the main characteristics of the model, the effect of scale in the hydrological response cannot be evaluated.

*Response: We will add more peer reviewed references of the W3RA model to the manuscript (Van Dijk, 2010b; Van Dijk and Renzullo, 2011; Van Dijk et al., 2012a). The main evaluation of the model on global scale is documented in Van Dijk (2013) in a Water Resources paper. The main description of the model code and parameters is given here (Van Dijk, 2010):*
http://www.clw.csiro.au/publications/waterforahealthycountry/2010/wfhc-aus-water-resources-assessment-system.pdf
*The model code is also open source and online available on Github:*
https://github.com/openstreams/wflow/blob/master/wflow-py/wflow/wflow_w3ra.py

*W3RA is a global model and therefore not specifically calibrated/tested for the Rhine and Mississippi regions. The model is indeed a semi-distributed conceptual model and it distinguishes between deep and shallow-rooting vegetation to represent the ecosystems (as mentioned in line 5 on page 5 in the manuscript). For an extensive description we refer to the documentation of the model, which presents all equations and parameters (see document above; Van Dijk, 2010). Furthermore, we would like to indicate more studies which use W3RA for analysing discharge at catchment scale (van Dijk, 2014; Beck et al., 2017; Schellekens et al 2017; van Dijk 2013), from which van Dijk (2014) and Beck et al. (2017) are published in HESS. We will edit the manuscript to ensure that the full description of the model and its source code is retrievable from the cited references.*

*Adaptions made: We have added more peer reviewed references of the W3RA model to the manuscript (Van Dijk, 2010b; Van Dijk and Renzullo, 2011; Van Dijk et al., 2012a). The main evaluation of the model on global scale is documented in Van Dijk (2013) in a Water Resources paper. The main description of the model code and parameters is given here (Van Dijk, 2010):*
http://www.clw.csiro.au/publications/waterforahealthycountry/2010/wfhc-aus-water-resources-assessment-system.pdf
*The model code is also open source and online available on Github:*
https://github.com/openstreams/wflow/blob/master/wflow-py/wflow/wflow_w3ra.py
*We have included both links in the code and data availability statement.*

Reviewer: The most crucial parameter in any hydrological model is the storage capacity that the vegetation has created in the root zone. Surprisingly, this root zone storage capacity is independent on the soil parameters, because the vegetation adjusts the rooting depth to the soil, so as to create sufficient buffer capacity to overcome drought. This root zone storage capacity is scale-independent and directly connected to climate (Gao et al., 2014b), can be applied globally (Wang-Erlandsson et al., 2016) and locally

(Boer-Euser et al., 2016), outperforming traditional soil-based approaches. So instead of using soil information and rooting depth as the main (and highly unreliable) input to hydrological models, it would be much better –and much easier – to use the scale-independent climate-based root zone  storage capacity as the key input. I am curious to hear the opinion of the authors on this issue.

*Response: We thank the reviewer for his comment. We agree that the rooting depth and root-zone storage (and crop factors) play an important role in hydrological models. We agree that this novel approach deserves full attention in the global hydrological modelling and therefore should be studied further. Application and testing of approaches developed by Wang-Erlandsson et al (2016) and others, or going to even more dynamical root density approaches (e.g. van Wijk and Bouten, 2001), is very interesting, in particular because of the scale independence . When reading the comments we realize we have not given full attention to these innovating topics on hydrological modelling and we will elaborate more on them in the discussion and suggest further work on it.*

*Adaptions made: We have added the potential use of root-zone storage capacity to the discussion section 5.2 on scale interaction in global hydrological modelling. :*
*'Different approaches are taken to develop models that are robust over a range of spatial scales for the application in global hydrological modelling. For example, several studies apply parameter regionalization techniques to adapt parameters across scales (Samaniego, 2010). Gao et al. (2016, 2013) suggest that, when physical processes are connected to the correct landscape-indicators, regionalisation can be applied without further calibration. The root-zone storage capacity is such a scale-independent parameter (Gao et al., 2014). This approach has a potential for resolution comparison studies, and it can be applied globally (Wang-Erlandsson et al., 2016).'*

Reviewer: My assessment:
I do consider this a well-written and well-prepared paper, but I find the research question not very innovative: testing with an ill-described model (probably developed for local circumstances) whether a reduction of resolution results in better performance. Runoff processes are largely determined by climate, ecosystem and topography. A model that requires calibration of scale-dependent parameters is not suitable for such an exercise. Although I support the conclusion about the reduction of scale in the meteorology and the difference between the dominant rainfall bringing mechanisms in the Mississippi and the Rhine, I doubt the adequacy of the study on reducing the scale of the hydrological model. The authors apparently missed a considerable part of the debate on hydrological modelling, as for instance presented in Hrachowitz et al. (2013), where these issues were summarized after an intensive debate among the entire hydrological community.

*Response: We thank prof. Savenije for the assessment. We are aware of the ongoing debate about scales and hydrological modelling, and we have sharpened our knowledge on root-zone storage capacity. We would like to emphasize that we do not claim that the approach taken here is the best. We have tested from a 'global modelling perspective' if scaling up the resolution  will lead to better performance (going from 0.5 to 0.05 degrees). We have investigated if rescaled parameters, and only using high-resolution information we are certain about (topography and vegetation), at higher resolution would give better simulation results. This also allows comparing the outcome of the models in a fair and transparent way, which would not be possible or be very difficult otherwise, because the model itself would change. As concluded in the manuscript, we find that the improvements are limited and that likely other process representation (e.g. subsurface lateral flow) is needed, especially if we move to even finer resolution than 0.05 degrees.*

*The comments of Huub Savenije stress the challenges in the field of hydrological modelling, especially at different scales. We hope this manuscript can contribute to this discussion by showing the results of one approach to deal with different spatial scales, which we conclude is not the best method. We will include an extra paragraph in the*

*discussion to put our approach in perspective with the challenges in hydrological modelling across scales, and the potential benefits of a different viewpoint, illustrated along the suggested references by the reviewer.*

*Adaptions made: We have added an extra section to the discussion to touch upon the issues raised by the reviewer. This new section 5.2 is called: Process presentation vs. increased resolution of the global hydrological model. In this section, we discuss multiple approaches to deal with scale interactions in global hydrology, like regional parameterizations or using scale-independent parameters, the latter which was suggested here. Thereafter we indicate that the approach of this study is to see if only increasing the resolution (and only the parameters we are sure of) leads to improved discharge. We find that this is not the case and therefore we can conclude that simply increasing the resolution is not the best way forward and that there is a need for better process presentation.*

*'We believe that different approaches to deal with scale interactions, among the ones we mention here, deserve rigorous study. In this study, we followed probably the most simple approach, namely testing from a 'global modelling perspective' if enhancing the resolution of the GHM will improve modelled discharge (going from 0.5° to 0.05°). We simply remapped the parameters from the low 0.5° to the high 0.05° resolution model, except for vegetation and orography which are known at high resolution (see Sect. methodology W3RA). This allows comparing the outcome of the models in a transparent way, which would not be possible or be very difficult otherwise, because the model itself would change. With this technique, we find no consistent improvements in discharge with the higher resolution GHM. This could be related to the choice of our GHM, as we only use one model, or to the need for more elaborated process representation (e.g. subsurface lateral flow). Our results therefore indicate that only increasing the resolution of the GHM has limited effect on simulating discharge. This conclusion is in line with the achievements/challenges of prediction in ungauged basins as summarized in Hrachowitz et al. (2013).'*

**Reviewer 3**

*We thank reviewer 3 for the constructive comments. We reply to the raised points below in italic, these are the replies from the open discussion. The replies in blue are additional to the open discussion, and indicate the changes made to the manuscript.*

Reviewer: I thank the authors for this revised version and for addressing my comments. In this revised version, the authors recognise that an increase in resolution does not necessarily lead to better streamflow simulations, and clearly state it in the abstract and in the main text. This made me wonder why it was not the case, so I went through the different versions of the manuscript, reviewers' comments and authors' replies. In a nutshell, I am concerned that i) the GHM was not ideally set up and ii) the experimental setup chosen by the authors prevented them from fully exploiting the benefits of the higher resolution.

GHM setup: When forced using ERAI, the GHM severely underestimates streamflow in the Rhine basin (Figure 5c), in particular in summer, when as much as half of the streamflow is missing. The authors recognise that the "GHM is too dry in the summer months", since ERAI precipitation estimates are "good". I raised this issue in a previous round of revisions, and the authors replied that they "have not performed an in-depth analysis on the performance of W3RA as this study focuses on the sensitivity to resolution." I regret to say that I do not agree with this argument. Before increasing the resolution, the authors should have made sure that the basic hydrological behaviour of the basin (its water balance) is adequately captured. In this specific case, the streamflow underestimation in summer indicates that the GHM is not adequately setup for the Rhine catchment.

*Response: In our analysis, we found comparable simulated discharge with ERAI for the Rhine as Photiadou et al (2011), who used a high resolution HBV96 calibrated hydrological model for the Rhine forced with ERAI. In contrast to their claims of underestimated precipitation, we found that the total precipitation over the Rhine catchment in ERAI is comparable with the observations. Due to resolution issues the distribution of the rainfall over the catchment might not be optimal, which could be important for snow build up in the Alps and hence could cause too low discharges in spring/summer. We found that the total actual evaporation over the Rhine basin is overestimated when compared to GLEAM. This could further contribute to the simulation results we obtained. However, the observed precipitation, and in particular evaporation contain large uncertainties (Bosilovich, 2008 ; Miralles, 2011). In our manuscript, we present a transparent and fair assessment of the mismatch between model and data. We could conduct a more in depth analysis of this issue. However, in our opinion, this will hardly contribute to the main aim of this work, which is testing the effect of resolution on simulating the hydrological cycle. We leave it up to the editor to decide if additional analysis is needed.*

*Adaptions made: the editor Stan Schymanski has indicated that no more extra analysis are needed to convey the message of this manuscript and therefore we have not addressed this concern of the reviewer.*

Reviewer: From a more general perspective, the authors mentioned in their reply that "The global hydrological model which we use is not calibrated for the specific catchments of this study. In general, calibration of global hydrological models is limited." I recognise that, when run at the global scale, GHMs are challenging to calibrate, but for this study, I consider that the authors should have taken the time to adjust model parameters in order to provide acceptable streamflow simulations for the two basins they selected, before increasing the GHM resolution.

*Response: This study focuses on the effect of increased resolution in simulating the hydrological cycle of river basins. Therefore, we use a global climate model and a global hydrological model. We examine our hypothesis by focussing on two large river basins, with different characteristics. To decrease the computing time we only run the global hydrological model for the two selected basins. However, we still do this from the viewpoint of 'global hydrology', i.e. using a global hydrological model not specifically designed for the two basins. If we want to obtain the best results in modelling the hydrological basins over the Rhine and Mississippi basin, we should have chosen specific regional models build for the specific basins, which is not the research question of the study. We intend to add an extra paragraph on the issue on global modelling and calibrated catchment scale modelling in response to reviewer 3.*

*Adaptions made: To emphasize in the manuscript that the goal of this study is not to get the best discharge results but to study the effect of increased resolution from a global modelling perspective, we have added a paragraph to the discussion section 5.2: 'Besides the discussion on process presentation versus high-resolution modelling, we would like to note that we have performed the simulations with W3RA from the viewpoint of 'global hydrology', i.e. using a global hydrological model not specifically designed for the two basins. Often global hydrological models are used in climate context, while not being optimized for individual basins. If we want to obtain the best results in modelling the hydrological balance over the Rhine and Mississippi basin, we should have chosen specific regional calibrated models build for the specific basins (calibrated-catchment models), which is not in line with our viewpoint here.'*

Reviewer: Parameter estimation under higher resolution: In the revised manuscript, the authors mention that for the GHM, "to allow a fair comparison between the two model resolutions, we remapped these parameters from the 0.5_ to the 0.05_ resolution." The authors explained in a previous round of reviews that "we remap the parameters from the low to the high resolution using the resample statement in PCRaster (Karssenberg et al., 2010)". How the remapping exactly works is unfortunately not documented in the revised manuscript, but my impression is that a resampling routine alone is not sufficient to incorporate the new data and knowledge necessary to enable the model to resolve smaller-scale processes. In their revised abstract, the authors report that "Increasing the resolution of vegetation and orography in the high resolution GHM (from 0.5_ to 0.05_) shows no significant differences in discharge for both basins, likely because the hydrological processes depend highly on model parameter values that are not readily available at high resolution." All this indicates that although the GHM was run at higher resolution, it was probably run using parameters at an effective resolution close to the original (coarser) resolution, because model parameters are higher resolution were not "readily available". This likely explains why the benefits of increasing the GHM resolution were limited. Arguably, increasing the resolution of a model goes beyond decreasing its grid spacing, it should also involve the incorporation of new data and knowledge on process representation across scales. It is my impression that this part is essentially missing and this is preventing us from truly assessing the benefits of the increased resolution.

*Response: In every version of the manuscript we mentioned "to allow a fair comparison between the two model resolutions, we remapped these parameters from the 0.5 to the 0.05 degrees resolution." Apparently, this was not clear enough and we will adjust the text to make this procedure clearer.*
*The reviewer understood correctly that in this study we only tested the impact of increased resolution by increasing resolution and not by including more/different smaller-scale processes or involving new data. We decided to do so, since applying all suggested adaptions to the high-resolution version would imply a different model which is not directly comparable with the low resolution version. Therefore, we investigated if rescaling parameters, and only using known high-resolution parameters (topography and vegetation), at higher resolution would give better results. It was shown by Melsen et al*

*(2016) that parameters can be transferred across the spatial scales, on regional scales from 1 km² to 100 km², and our work could be seen as a large-scale test of their work. We conclude our paper by stating exactly the view of the reviewer: quick gains are not straightforwardly achieved (line number 9-10-11 on page 27).*

*Adaptions made:*
*The resample statement of PCRaster applies an area-weighted interpolation technique to remap the parameters from 0.5 to 0.05 degrees. We have changed the manuscript accordingly: 'To allow a fair comparison between the two model resolutions, we remapped these parameters from the 0.5° to the 0.05° resolution using area weighted interpolation.'*
*We have added an additional section to the discussion to touch upon the issues raised by you and Huub Savenije. The section is called Process presentation vs. increased resolution of the global hydrological model.*

*In this section, we discuss multiple approaches to deal with scale interactions in global hydrology, like regional parameterizations or using scale-independent parameters. Thereafter we indicate that the approach of this study is to see if only increasing the resolution (and only the parameters we are sure of) leads to improved discharge. We find that this is not the case and therefore we can conclude that simply increasing the resolution is not the best way forward. By showing these results we hope to contribute to the larger discussion on scale interaction in global hydrology.*

[revised manuscript text omitted]

---

## Author Response (AR2)

Dear Authors,

Thank you for addressing the various comments and concerns raised by the referees. I think that due to the different perspectives added by the referees and your open discussion of these perspectives in the manuscript, the paper will be a very interesting and valuable addition to the scientific literature. I have not found an acknowledgement section. Given all the helpful and constructive comments submitted by the referees, I think it would be fair to acknowledge their input. You might also like to acknowledge your funding sources, so I am marking this as "accept subject to technical corrections" in order to give you the opporunity to do so.

Best regards,
Stan

Dear Stan Schymanski,

We agree that due the reviewers comments the manuscript improved substantially and added extra perspective to this study. We have now included an acknowledgement section where we thank the reviewers for their input and where we also acknowledge my funding sources.

Best regards,

Imme Benedict, on behalf of all co-authors